# The population genetics of collateral resistance and sensitivity

Sarah M Ardell, Sergey Kryazhimskiy*

Division of Biological Sciences, University of California, San Diego, La Jolla, United States

**Abstract** Resistance mutations against one drug can elicit collateral sensitivity against other drugs. Multi-drug treatments exploiting such trade-offs can help slow down the evolution of resistance. However, if mutations with diverse collateral effects are available, a treated population may evolve either collateral sensitivity or collateral resistance. How to design treatments robust to such uncertainty is unclear. We show that many resistance mutations in *Escherichia coli* against various antibiotics indeed have diverse collateral effects. We propose to characterize such diversity with a joint distribution of fitness effects (JDFE) and develop a theory for describing and predicting collateral evolution based on simple statistics of the JDFE. We show how to robustly rank drug pairs to minimize the risk of collateral resistance and how to estimate JDFEs. In addition to practical applications, these results have implications for our understanding of evolution in variable environments.

## Editor's evaluation

When selecting for one particular trait, it is not uncommon for other traits to change. This is due to pleiotropic mutations that affect multiple characters. Ardell and Kryazhimskiy develop a theoretical framework to predict adaptive trajectories observed in environments other than the one selection is operating in. The effects of adaptation across environments have important implication to antibiotic treatments, where resistance evolution to one antibiotic can alter the susceptibility to other antibiotics.

*For correspondence:
skryazhi@ucsd.edu

Competing interest: The authors declare that no competing interests exist.

## Introduction

The spread of resistance against most antibiotics and the difficulties in developing new ones has sparked considerable interest in using drug combinations and sequential drug treatments to treat bacterial infections, as well as cancers (*Pál et al., 2015*). Treatments where the drugs are chosen so that resistance against one of them causes the pathogen or cancer population to become sensitive to the other—a phenomenon known as collateral sensitivity—can eliminate the population before multi-drug resistance emerges (*Pál et al., 2015*; *Pluchino et al., 2012*).

The success of a multi-drug treatment hinges on knowing which drugs select for collateral sensitivity against which other drugs. This information is obtained empirically by exposing bacterial and cancer-cell populations to drugs and observing the evolutionary outcomes (*Roemhild et al., 2020*; *Jensen et al., 1997*; *Imamovic and Sommer, 2013*; *Lázár et al., 2018*; *Maltas and Wood, 2019*; *Batra et al., 2021*; *Sanz-García et al., 2020*; *Schenk et al., 2015*; *Lázár et al., 2013*; *Barbosa et al., 2019*; *Hernando-Amado et al., 2020*; *Kim et al., 2014*; *Jahn et al., 2021*; *Kavanaugh et al., 2020*; *Laborda et al., 2021*; *Oz et al., 2014*; *Munck et al., 2014*). Prior studies have largely focused on various empirical questions related to the evolution of collateral sensitivity and resistance, such as identifying their genetic basis (*Lázár et al., 2014*; *Roemhild et al., 2020*; *Maltas and Wood, 2019*; *Hernando-Amado et al., 2020*; *Kavanaugh et al., 2020*; *Laborda et al., 2021*), understanding how

**eLife digest** Drugs known as antibiotics are the main treatment for most serious infections caused by bacteria. However, many bacteria are acquiring genetic mutations that make them resistant to the effects of one or more types of antibiotics, making them harder to eliminate.

One way to tackle drug-resistant bacteria is to develop new types of antibiotics; however, in recent years, the rate at which new antibiotics have become available has been dwindling. Using two or more existing drugs, one after another, can also be an effective way to eliminate resistant bacteria. The success of any such 'multi-drug' treatment lies in being able to predict whether mutations that make the bacteria resistant to one drug simultaneously make it sensitive to another, a phenomenon known as collateral sensitivity.

Different resistance mutations may have different collateral effects: some may increase the bacteria's sensitivity to the second drug, while others might make the bacteria more resistant. However, it is currently unclear how to design robust multi-drug treatments that take this diversity of collateral effects into account.

Here, Ardell and Kryazhimskiy used a concept called JDFE (short for the joint distribution of fitness effects) to describe the diversity of collateral effects in a population of bacteria exposed to a single drug. This information was then used to mathematically model how collateral effects evolved in the population over time.

Ardell and Kryazhimskiy showed that this approach can predict how likely a population is to become collaterally sensitive or collaterally resistant to a second antibiotic. Drug pairs can then be ranked according to the risk of collateral resistance emerging, so long as information on the variety of resistance mutations available to the bacteria are included in the model.

Each year, more than 700,000 people die from infections caused by bacteria that are resistant to one or more antibiotics. The findings of Ardell and Kryazhimskiy may eventually help clinicians design multi-drug treatments that effectively eliminate bacterial infections and help to prevent more bacteria from evolving resistance to antibiotics. However, to achieve this goal, more research is needed to fully understand the range collateral effects caused by resistance mutations.

collateral outcomes depend on treatment design (e.g. sequential versus combination) (*Lázár et al., 2014*; *Munck et al., 2014*; *Bergstrom et al., 2004*; *Batra et al., 2021*; *Sanz-García et al., 2020*; *Schenk et al., 2015*; *Lázár et al., 2013*; *Kim et al., 2014*; *Jahn et al., 2021*), or testing whether collateral sensitivity is an evolutionarily stable outcome (*Barbosa et al., 2019*). However, one important feature of these experimental studies has received little attention, namely, the fact that different experiments often produce collateral sensitivity profiles that are inconsistent with each other (e.g. *Imamovic and Sommer, 2013*; *Oz et al., 2014*; *Barbosa et al., 2017*; *Maltas and Wood, 2019*). Some inconsistencies can be explained by the fact that resistance mutations vary between bacterial strains, drug dosages, etc. (*Mira et al., 2015*; *Barbosa et al., 2017*; *Das et al., 2020*; *Pinheiro et al., 2021*; *Card et al., 2021*; *Gjini and Wood, 2021*). However, wide variation in collateral outcomes is observed even between replicate populations (*Oz et al., 2014*; *Barbosa et al., 2017*; *Maltas and Wood, 2019*; *Nichol et al., 2019*). This variation suggests that bacteria and cancers have access to multiple resistance mutations with diverse collateral effects and that replicate populations accumulate different resistance mutations due to the intrinsic randomness of the evolutionary process (*Jerison et al., 2020*; *Nichol et al., 2019*). However, the diversity of collateral effects among resistance mutations has rarely if ever been systematically measured. Moreover, few existing approaches for designing robust multi-drug treatments have modelled this mutational diversity explicitly within the population genetics context (*Nichol et al., 2019*; *Maltas and Wood, 2019*). Yet, a theory grounded in population genetics could help us understand how the expected collateral outcomes and the uncertainty around these expectations depend on evolutionary parameters and how these expectations and uncertainties change over time.

Here, we develop such a theory. Collateral sensitivity and resistance are specific examples of the more general evolutionary phenomenon, pleiotropy, which refers to any situation when one mutation affects multiple phenotypes (*Wagner and Zhang, 2011*; *Paaby and Rockman, 2013*). In case of drug resistance evolution, the direct effect of resistance mutations is to increase fitness in the presence of

one drug (the 'home' environment). In addition, they may also provide pleiotropic gains or losses in fitness in the presence of other drugs (the 'non-home' environments) leading to collateral resistance or sensitivity, respectively.

Classical theoretical work on pleiotropy has been done in the field of quantitative genetics (*Lande and Arnold, 1983*; *Rose, 1982*; *Barton, 1990*; *Slatkin and Frank, 1990*; *Jones et al., 2003*; *Johnson and Barton, 2005*). In these models, primarily developed to understand how polygenic traits respond to selection in sexual populations, pleiotropy manifests itself as a correlated temporal change in multiple traits in a given environment. The question of how new strongly beneficial mutations that accumulate in asexual populations evolving in one environment affect its fitness in future environments is outside of the scope of these models.

The pleiotropic consequences of adaptation have also been explored in various 'fitness landscape' models (e.g. *Connallon and Clark, 2015*; *Martin and Lenormand, 2015*; *Harmand et al., 2017*; *Wang and Dai, 2019*; *Maltas et al., 2021*; *Nichol et al., 2019*; *Tikhonov et al., 2020*). In particular, *Nichol et al., 2019* specifically addressed the problem of diversity of collateral resistance/sensitivity outcomes in the context of a combinatorially complete fitness landscapes of four mutations in the TEM β-lactamase gene. They found that different *in silico* populations adapting to the same antibiotic often arrive at different fitness peaks which results in different levels of collateral resistance/sensitivity against other drugs. They observed qualitatively similar variability in the collateral outcomes among replicate populations of the bacterium *Escherichia coli* evolving in the presence of cefotaxime (CTX), although it is unclear whether different populations indeed arrived at different fitness peaks. In general, the fitness landscape approach helps us understand how evolutionary trajectories and outcomes depend on the global structure of the underlying fitness landscape. However, applying this approach in practice is challenging because the global structure of fitness landscapes is unknown and notoriously difficult to estimate, even in controlled laboratory conditions.

Here, we take a different approach which is agnostic with respect to the global structure of the fitness landscape. Instead, we assume only the knowledge of the so-called joint distribution of fitness effects (JDFE), that is, the probability that a new mutation has a certain pair of fitness effects in the home and non-home environments (*Jerison et al., 2014*; *Martin and Lenormand, 2015*; *Bono et al., 2017*). The JDFE is a natural extension of the DFE, the distribution of fitness effects of new mutations, often used in modeling evolution in a single environment (*King, 1972*; *Ohta, 1987*; *Orr, 2003*; *Kassen and Bataillon, 2006*; *Eyre-Walker and Keightley, 2007*; *Martin and Lenormand, 2008*; *MacLean and Buckling, 2009*; *Kryazhimskiy et al., 2009*; *Levy et al., 2015*). Like the DFE, the JDFE is a local property of the fitness landscape which means that it can be, at least in principle, estimated by using a variety of modern high-throughput techniques (e.g. *Qian et al., 2012*; *van Opijnen et al., 2009*; *Chevereau et al., 2015*; *Levy et al., 2015*; *Blundell et al., 2019*; *Aggeli et al., 2021*). The downside of this approach is that the JDFE can change over time as the population traverses the fitness landscape (*Good et al., 2017*; *Venkataram et al., 2020*; *Aggeli et al., 2021*). However, in the context of collateral drug resistance and sensitivity, we are primarily interested in short time scales over which the JDFE can be reasonably expected to stay approximately constant.

The rest of the paper is structured as follows. First, we use previously published data to demonstrate that *E. coli* has access to drug resistance mutations with diverse collateral effects. This implies that, rather than treating collateral effects as deterministic properties of drug pairs, we should think of them probabilistically, in terms of the respective JDFEs. We then show that a naive intuition about how the JDFE determines pleiotropic outcomes of evolution can sometimes fail, and a mathematical model is therefore required. We develop such a model, which reveals two key 'pleiotropy statistics' of the JDFE that determine the dynamics of fitness in the non-home condition. Our theory makes quantitative predictions in a variety of regimes if the population genetic parameters are known. However, we argue that in the case of drug resistance evolution the more important problem is to robustly order drug pairs in terms of their collateral sensitivity profiles even if the population genetic parameters are unknown. We develop a metric that allows us to do so. Finally, we provide some practical guidance for estimating the pleiotropy statistics of empirical JDFEs in the context of ranking drug pairs.

## Results

### Antibiotic resistance mutations in *E. coli* have diverse collateral effects

We begin by demonstrating that the JDFE is a useful concept for modeling the evolution of collateral antibiotic resistance and sensitivity. If all resistance mutations against a given drug had identical pleiotropic effects on the fitness of the organism in the presence of another drug, the dynamics of collateral resistance/sensitivity could be understood without the JDFE concept. On the other hand, if different resistance mutations have different pleiotropic fitness effects, predicting the collateral resistance/sensitivity dynamics requires specifying the probabilities with which mutations with various home and non-home fitness effects arise in the population. The JDFE specifies these probabilities. Therefore, for the JDFE concept to be useful in the context of collateral resistance/sensitivity evolution, we need to show that resistance mutations against common drugs have diverse collateral effects in the presence of other drugs.

To our knowledge, no data sets are currently publicly available that would allow us to systematically explore the diversity of collateral effects among all resistance mutations against any one drug in any organism. Instead, we examined the fitness effects of 3883 gene knock-out mutations in the bacterium *Escherichia coli*, measured in the presence of six antibiotics (*Chevereau et al., 2015*), as well as the fitness effects of 4997 point mutations in the TEM-1 β-lactamase gene measured in the presence of two antibiotics (*Stiffler et al., 2015*).

For four out of six antibiotics used by *Chevereau et al., 2015*, we find between 12 (0.31 %) and 170 (4.38%) knock-out mutations that provide some level of resistance against at least one of the antibiotics (false discovery rate (FDR) ~25%; *Figure 1*, *Figure 1—source data 1*; see Materials and methods for details). Plotting on the $x$-axis the fitness effect of each knock-out mutation in the presence of the drug assumed to be applied first (i.e. the home environment) against its effect in the presence of another drug assumed to be applied later (i.e. the non-home environment, $y$-axis), we find mutations in all four quadrants of this plane, for all 12 ordered drug pairs (*Figure 1*, *Figure 1—source data 1*). Similarly, we find diverse collateral effects among mutations within a single gene (*Figure 1—figure supplement 1*; see Materials and methods for details).

Since both data sets represent subsets of all resistance mutations, we conclude that *E. coli* likely have access to resistance mutations with diverse pleiotropic effects, such that a fitness gain in the presence of any one drug can come either with a pleiotropic gain or a pleiotropic loss of fitness in the presence of other drugs. Therefore, the JDFE framework is suitable for modeling the evolution of collateral resistance/sensitivity. In the next section, we formally define a JDFE and probe our intuition for how its shape determines the fitness trajectories in the non-home environment.

### JDFE determines the pleiotropic outcomes of adaptation

For any genotype $g$ that finds itself in one ('home') environment and may in the future encounter another 'non-home' environment, we define the JDFE as the probability density $\Phi_g\left(\Delta x, \Delta y\right)$ that a new mutation that arises in this genotype has the selection coefficient $\Delta x$ in the home environment and the selection coefficient $\Delta y$ in the non-home environment (*Jerison et al., 2014*). For concreteness, we define the fitness of a genotype as its Malthusian parameter (*Crow and Kimura, 1972*). So, if the home and non-home fitness of genotype $g$ are $x$ and $y$, respectively, and if this genotype acquires a mutation with selection coefficients $\Delta x$ and $\Delta y$, its fitness becomes $x + \Delta x$ and $y + \Delta y$. This definition of the JDFE can, of course, be naturally extended to multiple non-home environments. In principle, the JDFE can vary from one genotype to another. However, to develop a basic intuition for how the JDFE determines pleiotropic outcomes, we assume that all genotypes have the same JDFE. We discuss a possible extension to epistatic JDFEs in Appendix 1.

The JDFE is a complex object. So, we first ask whether some simple and intuitive summary statistics of the JDFE may be sufficient to predict the dynamics of the non-home fitness of a population that is adapting in the home environment. Intuitively, if there is a trade-off between home and non-home fitness, non-home fitness should decline; if the opposite is true, non-home fitness should increase. Canonically, a trade-off occurs when any mutation that improves fitness in one environment decreases it in the other environment and vice versa (*Roff and Fairbairn, 2007*). Genotypes that experience such 'hard' trade-offs are at the Pareto front (*Shoval et al., 2012*; *Li et al., 2019*). For genotypes that are not at the Pareto front, some mutations that are beneficial in the home environment may be beneficial in the non-home environment and others may be deleterious. In this more general case,

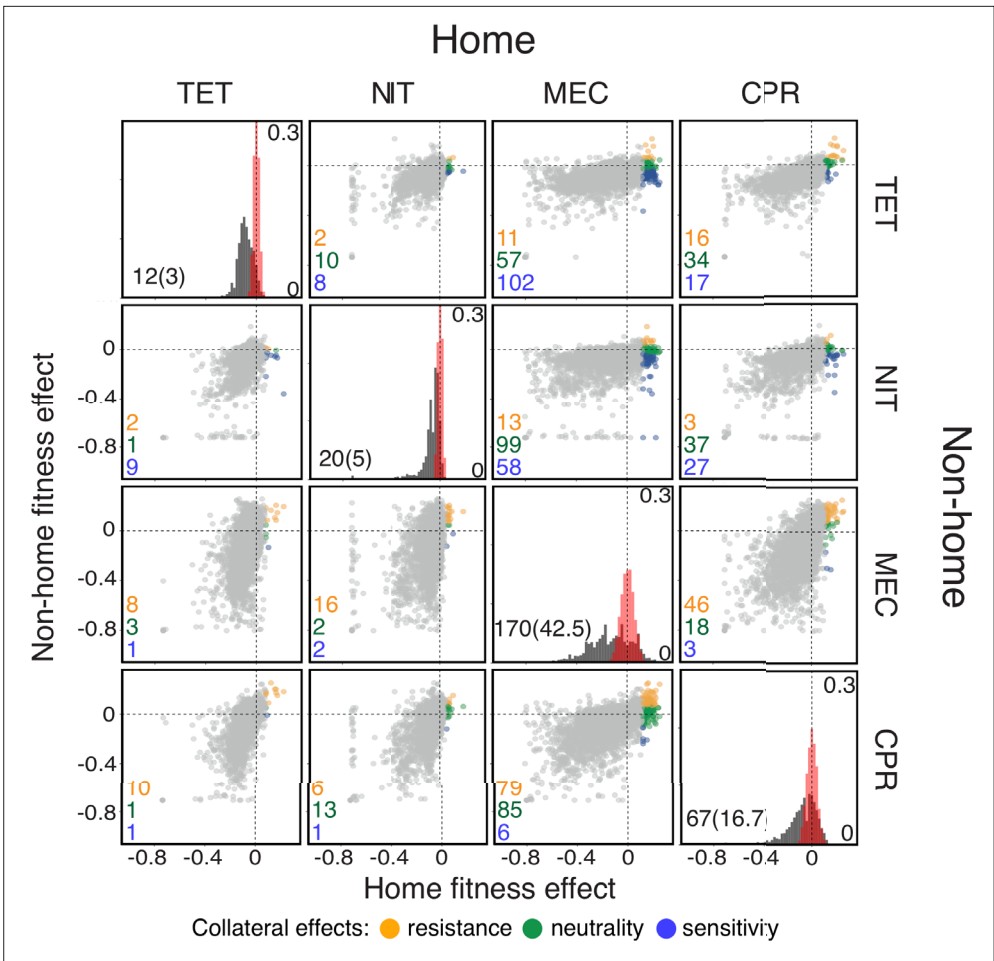

**Figure 1.** Fitness effects of gene knock-out mutations in *E. coli* in the presence of four antibiotics. Data are from *Chevereau et al., 2015*. Each diagonal panel shows the distribution of fitness effects (DFE) of knock-out mutations in the presence of the corresponding antibiotic (equivalent to Figure 1C in *Chevereau et al., 2015*). Scale of the *y*-axis in these panels is indicated inside on the right. The estimated measurement noise distributions are shown in red (see Materials and methods for details). Note that some noise distributions are vertically cut-off for visual convenience. The number of identified beneficial mutations (i.e. resistance mutations) and the expected number of false positives (in parenthesis) are shown in the bottom left corner. The list of identified resistance mutations is given in the *Figure 1—source data 1*. Off-diagonal panels show the fitness effects of knock-out mutations across pairs of drug environments. The *x*-axis shows the fitness in the environment where selection would happen first (i.e., the 'home' environment). Each point corresponds to an individual knock-out mutation. Resistance mutations identified in the home environment are colored according to their collateral effects, as indicated in the legend. The numbers of mutations of each type are shown in the corresponding colors in the bottom left corner of each panel. TET: tetracycline; NIT: nitrofurantoin; MEC: mecillinam; CPR: ciprofloxacin.

The online version of this article includes the following source data and figure supplement(s) for figure 1:

**Figure supplement 1.** JDFEs of single point mutations in TEM-1 β-lactamase gene in *E. coli* in the presence of cefotaxime and ampicillin.

**Source data 1.** P-values and calls of collateral effects of beneficial knock-out mutations in the *Chevereau et al., 2015* data (see Materials and methods for details).

**Source data 2.** Calls of collateral effects of mutations beneficial in CTX in the *Stiffler et al., 2015* data (see Materials and methods for details).

trade-offs are commonly quantified by the degree of negative correlation between the effects of mutations on fitness in the two environments (*Roff and Fairbairn, 2007*; *Tikhonov et al., 2020*). Thus, we might expect that evolution on negatively correlated JDFEs would lead to pleiotropic fitness losses and evolution on positively correlated JDFEs would lead to pleiotropic fitness gains.

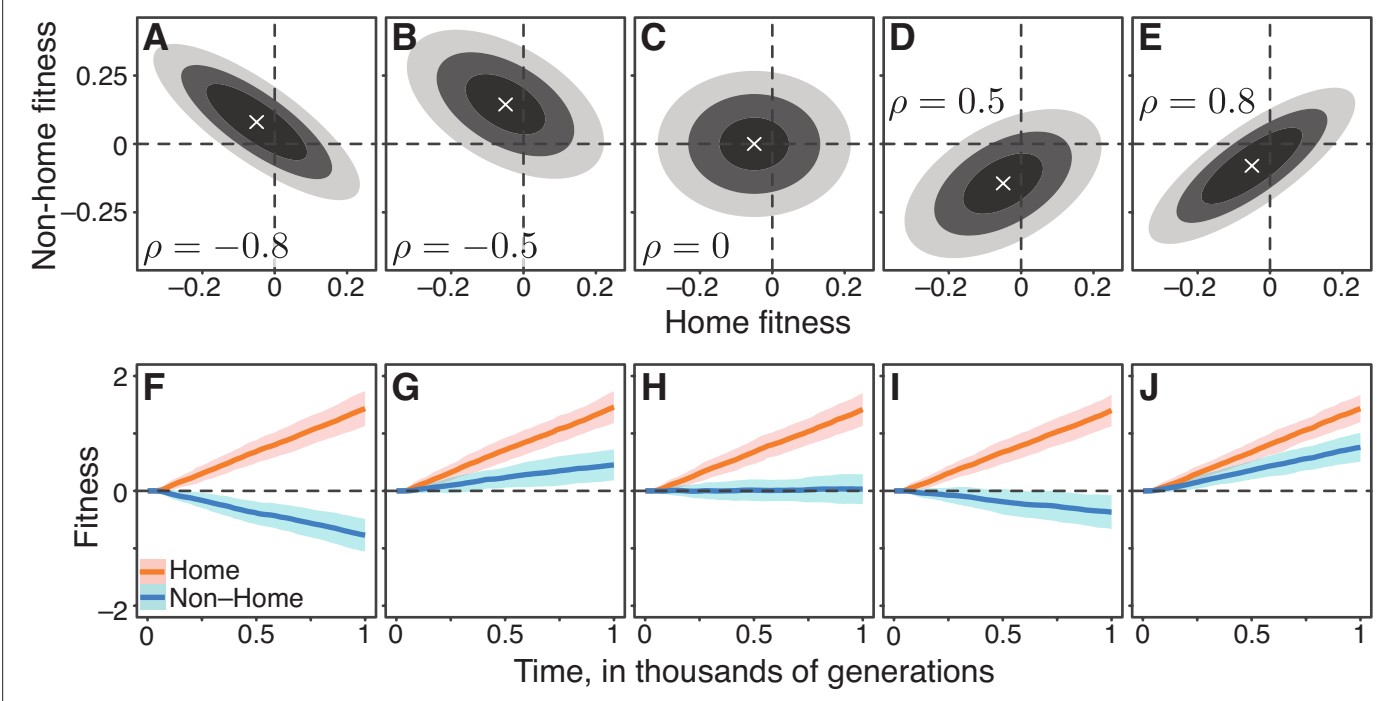

**Figure 2.** Gaussian JDFEs and the resulting fitness trajectories. (**A–E**) Contour lines for five Gaussian JDFEs. "x" marks the mean. For all distributions, the standard deviation is 0.1 in both home- and non-home environments. The correlation coefficient $\rho$ is shown in each panel. (**F–J**) Home and non-home fitness trajectories for the JDFEs shown in the corresponding panels above. Thick lines show the mean, ribbons show ±1 standard deviation estimated from 100 replicate simulations. Population size $N = 10^4$, mutation rate $U = 10^{-4}$ ($U_b = 4.6 \times 10^{-5}$).

The online version of this article includes the following figure supplement(s) for figure 2:

**Figure supplement 1.** JDFEs with equal probability weights in the first and fourth quadrants and the resulting fitness trajectories.

To test this intuition, we generated a family of Gaussian JDFEs that varied, among other things, by their correlation structure (**Figure 2**; Materials and methods). We then simulated the evolution of an asexual population on these JDFEs using a standard Wright-Fisher model (Materials and methods) and tested whether the trade-off strength, measured by the JDFE's correlation coefficient, predicts the dynamics of non-home fitness. **Figure 2** shows that our naive expectation is incorrect. Positively correlated JDFEs sometimes lead to pleiotropic fitness losses (**Figure 2D,I**), and negatively correlated JDFEs sometimes lead to pleiotropic fitness gains (**Figure 2B,G**). Even if we calculate the correlation coefficient only among mutations that are beneficial in the home environment, the pleiotropic outcomes still do not always conform to the naive expectation, as the sign of the correlation remains the same as for the full JDFEs in all these examples.

There are other properties of the JDFE that we might intuitively expect to be predictive of the pleiotropic outcomes of adaptation. For example, among the JDFEs considered in **Figure 2**, it is apparent that those with similar relative probability weights in the first and fourth quadrants produce similar pleiotropic outcomes. However, simulations with other JDFE shapes show that even distributions that are similar according to this metric can also result in qualitatively different pleiotropic outcomes (**Figure 2—figure supplement 1**).

Overall, our simulations show that JDFEs with apparently similar shapes can produce qualitatively different trajectories of pleiotropic fitness changes (e.g. compare **Figure 2A, F and B, G** or **Figure 2D,I and E,J**). Conversely, JDFEs with apparently different shapes can result in rather similar pleiotropic outcomes (e.g. compare **Figure 2B, G and E, J** or **Figure 2A,F and D,I**). Thus, while the overall shape of the JDFE clearly determines the trajectory of pleiotropic fitness changes, it is not immediately obvious what features of its shape play the most important role, particularly if the JDFE is more complex than a multivariate Gaussian. In other words, even if we have perfect knowledge of the fitness effects of all mutations in multiple environments, converting this knowledge into a qualitative prediction of the expected direction of pleiotropic fitness change (gain or loss) does not appear

straightforward. Therefore, we next turn to developing a population genetics model that would allow us to predict not only the direction of pleiotropic fitness change but also the expected rate of this change and the uncertainty around the expectation.

## The population genetics of pleiotropy

To systematically investigate which properties of the JDFE determine the pleiotropic fitness changes in the non-home environment, we consider a population of size $N$ that evolves on a JDFE in the 'strong selection weak mutation' (SSWM) regime, also known as the 'successional mutation' regime (*Orr, 2000*; *Desai and Fisher, 2007*; *Kryazhimskiy et al., 2009*; *Good and Desai, 2015*).

We consider an arbitrary JDFE without epistasis, that is a situation when all genotypes have the same JDFE $\Phi(\Delta x, \Delta y)$. We explore an extension to JDFEs with a simple form of epistasis in Appendix 1. We assume that mutations arise at rate $U$ per individual per generation. In the SSWM limit, a mutation that arises in the population either instantaneously fixes or instantaneously dies out. Therefore, the population is essentially monomorphic at all times, such that at any time $t$ we can characterize it by its current pair of fitness values $(X_t, Y_t)$. If a new mutation with a pair of selection coefficients $(\Delta x, \Delta y)$ arises in the population at time $t$, it fixes with probability $\pi(\Delta x) = \frac{1-e^{-2\Delta x}}{1-e^{-2N\Delta x}}$ (*Kimura, 1962*) in which case the population's fitness transitions to a new pair of values $(X_t + \Delta x, Y_t + \Delta y)$. If the mutation dies out, an event that occurs with probability $1 - \pi(\Delta x)$, the population's fitness does not change. This model specifies a continuous-time two-dimensional Markov process.

In general, the dynamics of the probability density $p(x, y, t)$ of observing the random vector $(X_t, Y_t)$ at values $(x, y)$ are governed by an integro-differential forward Kolmogorov equation, which is difficult to solve (Materials and methods). However, if most mutations that contribute to adaptation have small effects, these dynamics are well approximated by a diffusion equation which can be solved exactly (Materials and methods). Then $p(x, y, t)$ is a normal distribution with mean vector

$$\boldsymbol{m}(t) = \begin{pmatrix} x_0 \\ y_0 \end{pmatrix} + \begin{pmatrix} r_1 \\ r_2 \end{pmatrix} N U_b\, t \tag{1}$$

and variance-covariance matrix

$$\boldsymbol{\sigma}^2(t) = \begin{pmatrix} D_{11} & D_{12} \\ D_{12} & D_{22} \end{pmatrix} N U_b\, t, \tag{2}$$

where are $r_1$ and $r_2$, given by *Equation 7* and *Equation 8* in Materials and methods, are the expected fitness effects in the home and non-home environments for a mutation fixed in the home environment, and $D_{11}, D_{12}$, and $D_{22}$, given by *Equation 9*–*Equation 11* in Materials and methods, are the second moments of this distribution. Here, $U_b = U \int_{-\infty}^{\infty} d\eta \int_0^{\infty} d\xi \Phi(\xi, \eta)$ is the total rate of mutations beneficial in the home environment, and $x_0$ and $y_0$ are the initial values of population's fitness in the home and non-home environments.

*Equation 1* and *Equation 2* show that the distribution of population's fitness at time $t$ in the non-home environment is entirely determined by two parameters, $r_2$ and $D_{22}$, which we call the 'pleiotropy statistics' of the JDFE. The expected rate of fitness change in the non-home environment depends on the pleiotropy statistic $r_2$, which we refer to as the expected pleiotropic effect. Thus, evolution on a JDFE with a positive $r_2$ is expected to result in pleiotropic fitness gains and evolution on a JDFE with a negative $r_2$ is expected to result in pleiotropic fitness losses. *Equation 2* shows that the variance around this expectation is determined by the pleiotropy variance statistic $D_{22}$. Since both the expectation and the variance change linearly with time (provided $r_2 \neq 0$), the change in the non-home fitness in any replicate population will eventually have the same sign as $r_2$, but the time scale of such convergence depends on the 'collateral resistance risk' statistic $c = r_2/\sqrt{D_{22}}$ (Materials and methods). This observation has important practical implications, and we return to it in the Section 'Robust ranking of drug pairs'.

These theoretical results suggest a simple explanation for the somewhat counter-intuitive observations in *Figure 2*. We may intuitively believe that evolution on negatively correlated JDFEs should lead to fitness losses in the non-home environment because on such JDFEs mutations with largest fitness benefits in the home environment typically have negative pleiotropic effects. However, such mutations

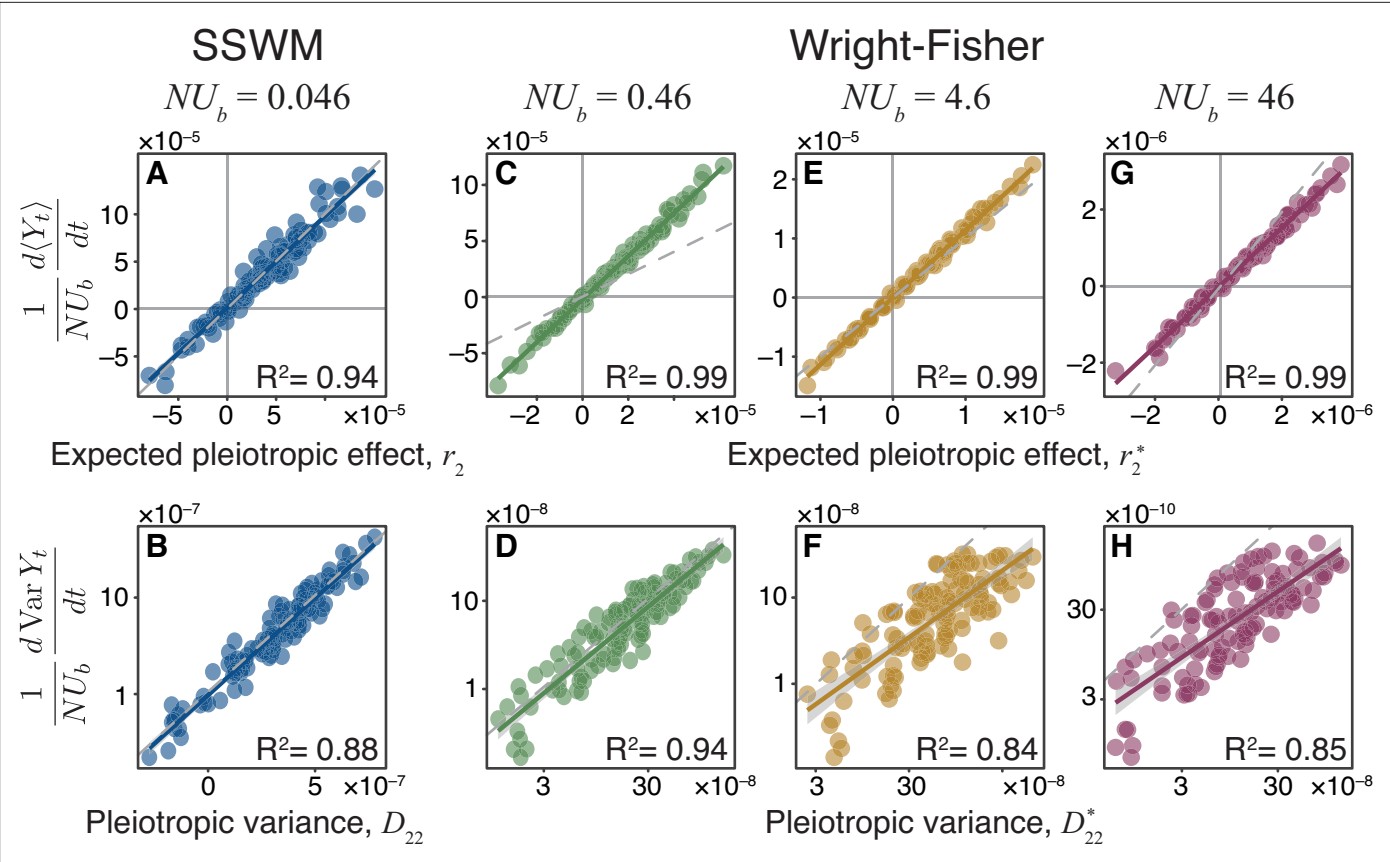

**Figure 3.** Pleiotropy statistics predict the properties of non-home fitness trajectories in simulations. Each point corresponds to an ensemble of replicate simulation runs with the same population genetic parameters on one of 125 Gaussian JDFEs (see *Figure 3—source data 1* for the JDFE parameters). (**A**) Expected pleiotropic effect $r_2$ versus the scaled slope of the mean rate of non-home fitness change observed in SSWM simulations. (**B**) Pleiotropic variance $D_{22}$ versus the scaled rate of change in the variance in non-home fitness observed in SSWM simulations. (**C, E, G**) Expected pleiotropic effect $r_2^*$ versus the scaled slope of the mean rate of non-home fitness change observed in Wright-Fisher simulations. (**D, F, H**) Pleiotropic variance $D_{22}^*$ versus the scaled rate of change in the variance in non-home fitness observed in Wright-Fisher simulations. (See *Figure 3—figure supplement 1* for comparison between simulations and the unadjusted pleiotropy statistics $r_2$ and $D_{22}$) 1000 replicate simulations were carried out in the SSWM regime. All Wright-Fisher simulations were carried out with $U = 10^{-4}$ and variable $N$, 300 replicate simulations per data point (see Materials and methods for details). In all panels, the gray dashed line represents the identity (slope 1) line, and the solid line of the same color as the points is the linear regression for the displayed points ($R^2$ value is shown in each panel; $P < 2 \times 10^{-16}$ for all regressions).

The online version of this article includes the following source data and figure supplement(s) for figure 3:

**Source data 1.** Parameters and summary statistics of simulation results for all Gaussian JDFEs used in *Figure 3*.

**Figure supplement 1.** Same as *Figure 3C-H*, but with $r_2$ and $D_{22}$ shown on the x-axis.

may be too rare to drive adaptation. At the same time, the more common mutations that do typically drive adaptation may have positive pleiotropic effects, in which case the population would on average gain non-home fitness, as in *Figure 2B*. Our theory shows that to predict the direction of non-home fitness change, the frequency of beneficial mutations with various pleiotropic effects and the strength of these effects need to be weighted by the likelihood that these mutations fix. The expected pleiotropic effect $r_2$ accomplishes this weighting.

We tested the validity of *Equation 1* and *Equation 2* by simulating evolution in the SSWM regime on 125 Gaussian JDFEs with various parameters (Materials and methods) and found excellent agreement (*Figure 3A and B*). However, many microbes likely evolve in the 'concurrent mutation' regime, that is, when multiple beneficial mutations segregate in the population simultaneously (*Desai and Fisher, 2007*; *Lang et al., 2013*). As expected, our theory fails to quantitatively predict the pleiotropic fitness trajectories when $NU_b > 1$ (*Figure 3—figure supplement 1*). However, the expected rate of change of non-home fitness and its variances remain surprisingly well correlated with the pleiotropy statistics $r_2$ and $D_{22}$ across various JDFEs (*Figure 3—figure supplement 1*). In other words, we can

still use these statistics to correctly predict whether a population would lose or gain fitness in the non-home environment and to order the non-home environments according to their expected pleiotropic fitness changes and variances. We will exploit the utility of such ranking in the next section.

We next sought to expand our theory to the concurrent mutation regime. A key characteristic of adaptation in this regime is that mutations whose fitness benefits in the home environment are below a certain 'effective neutrality' threshold are usually outcompeted by superior mutations and therefore fix with lower probabilities than predicted by Kimura's formula (*Schiffels et al., 2011*; *Good et al., 2012*). *Good et al., 2012* provide an equation for calculating the fixation probability $\pi^*(\Delta x)$ for a mutation with home fitness benefit $\Delta x$ in the concurrent mutation regime (Equation (6) in *Good et al., 2012*). Thus, by replacing $2\xi$ (the approximate fixation probability in the SSWM regime) in *Equation 8* and *Equation 11* with $\pi^*(\xi)$, we obtain the adjusted pleiotropy statistics $r_2^*$ and $D_{22}^*$ for the concurrent mutation regime (see Materials and methods for details). Note that in contrast to $r_2$ and $D_{22}$, the adjusted statistics $r_2^*$ and $D_{22}^*$ depend on the population genetic parameters $N$ and $U_b$.

To test how well these statistics predict the dynamics of fitness in the non-home environment, we simulated evolution on the same 125 JDFEs using the full Wright-Fisher model with a range of population genetic parameters that span the transition from the successional to the concurrent mutation regimes for 1,000 generations. We find that $r_2^*$ quantitatively predicts the expected rate of non-home fitness change, with a similar accuracy as *Good et al., 2012* predict the rate of fitness change in the home environment, as long as $NU_b > 1$ (*Figure 3C, E and G*; compare with *Figure 3—figure supplement 1A,C,E*). $D_{22}^*$ also predicts the empirically observed variance in non-home fitness trajectories much better than $D_{22}$, although this relationship is more noisy than between mean fitness and $r_2^*$ (*Figure 3D, F and H*; compare with *Figure 3—figure supplement 1B,D,F*). Some of this noise can be attributed to sampling, as we estimate both the mean and the variance from 300 replicate simulation runs, and the variance estimation is more noisy. Even in the absence of sampling noise however, we do not expect that $D_{22}^*$ would predict the non-home fitness variance perfectly because our theory does not account for the autocorrelation in the fitness trajectories that arise in the concurrent mutation regime but not in the successive mutation regime (see Appendix D in *Desai and Fisher, 2007*). To our knowledge, a rigorous analytical calculation for ensemble variance in fitness even in the home environment is not yet available.

Overall, our theory allows us to quantitatively predict the dynamics of non-home fitness in a range of evolutionary regimes if the JDFE and the population genetic parameters $N$ and $U_b$ are known. However, neither the full JDFE nor the population genetic parameters will likely be known in most practical situations, such as designing a drug treatment for a cancer patient. In the next section, we address the question of how to robustly select drug pairs for a sequential treatment, assuming that the pleiotropy statistics $r_2$ and $D_{22}$ are known but the population genetic parameters are not. In the Section 'Measuring JDFEs', we provide some guidance on how the JDFE can be measured.

## Robust ranking of drug pairs

Consider a hypothetical scenario where a drug treatment is being designed for a patient with a tumor or a bacterial infection. In selecting a drug, it is desirable to take into account not only the standard medical considerations, such as drug availability, toxicity, etc., but also the possibility that the treatment with this drug will fail due to the evolution of resistance. Therefore, it may be prudent to consider a list of drugs pairs (or higher-order combinations), ranked by the propensity of the first drug in the pair to elicit collateral resistance against the second drug in the pair. All else being equal, the drug deployed first should form a high-ranking pair with at least one other secondary drug. Then, if the treatment with the first drug fails, a second one can be deployed with a minimal risk of collateral resistance. Thus, we set out to develop a metric for ranking drug pairs according to this risk.

Clearly, any drug pair with a negative $r_2$ is preferable over any drug pair with a positive $r_2$, since the evolution in the presence of the first drug in a pair with $r_2 < 0$ is expected to elicit collateral sensitivity against the second drug in the pair but the opposite is true for drug pairs with $r_2 > 0$. It is also clear that among two drug pairs with negative $r_2$, a pair with a more negative $r_2$ and lower $D_{22}$ is preferable over a pair with a less negative $r_2$ and higher $D_{22}$ because evolution in the presence of the first drug in the former pair will more reliably lead to stronger collateral sensitivity against the second drug in the pair. The difficulty is in how to compare and rank two drug pairs where one pair has a more negative $r_2$ but higher $D_{22}$. Our theory shows that the chance of emergence of collateral resistance monotonically

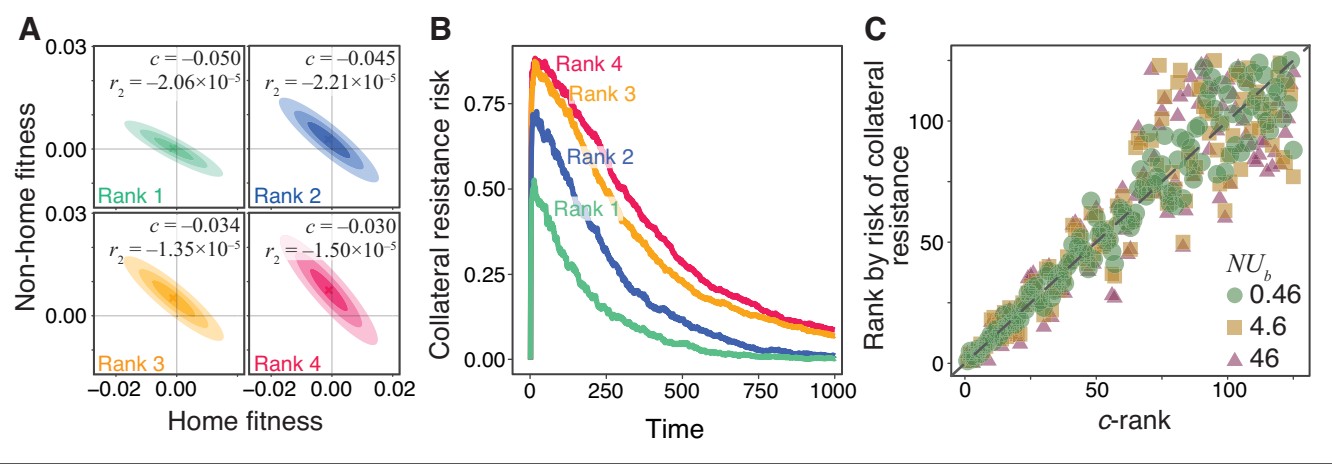

**Figure 4.** Robust ranking of drug pairs. (**A**) Four hypothetical JDFEs, ranked by their $c$ statistic. For all four JDFEs, the mean and the standard deviation in the home environment are $-1 \times 10^{-3}$ and $0.01$, respectively. The mean and the standard deviation in the non-home environment are $1 \times 10^{-4}$ and $5.1 \times 10^{-3}$ (rank 1), $2.6 \times 10^{-3}$ and $7.5 \times 10^{-3}$ (rank 2), $5.1 \times 10^{-3}$ and $7.5 \times 10^{-3}$ (rank 3), $7.5 \times 10^{-3}$ and $0.01$ (rank 4). Correlation coefficient for all four JDFEs is $-0.9$. (**B**) Collateral resistance risk over time, measured as the fraction of populations with positive mean fitness in the non-home environment. These fractions are estimated from 1000 replicate Wright-Fisher simulation runs with $N = 10^{4}$, $U = 10^{-4}$ ($NU_b = 0.46$). Colors correspond to the JDFEs in panel A. Numbers indicate the -rank of each JDFE. (**C**) A priori $c$-rank ($x$-axis) versus the a posteriori rank ($y$-axis) based on the risk of collateral resistance observed in simulations, for all 125 Gaussian JDFEs and all $NU_b$ values shown in **Figure 3**. Gray dashed line is the identity line. $R^2$ values are $0.94, 0.81$ and $0.82$ for $NU_b = 0.46, 4.6$ and 46, respectively. $P < 10^{-15}$ for all regressions.

The online version of this article includes the following figure supplement(s) for figure 4:

**Figure supplement 1.** Ranking of AMP concentrations according to their risk of collateral resistance, based on **Stiffler et al., 2015** data.

increases with the collateral risk statistic $c = r_2/\sqrt{D_{22}}$ (see Materials and methods). Thus, we propose to rank drug pairs by $c$ from lowest (most negative and therefore most preferred) to highest (least negative or most positive and therefore least preferred).

To demonstrate the utility of such ranking, consider four hypothetical drug pairs with JDFEs shown in **Figure 4A**. The similarity between their shapes makes it difficult to predict a priori which one would have the lowest and highest probabilities of collateral resistance. Thus, we rank these JDFEs by their $c$ statistic. To test whether this ranking is accurate with respect to the risk of collateral resistance, we simulate the evolution of a Wright-Fisher population in the presence of the first drug in each pair for 600 generations and estimate the probability that the evolved population has a positive fitness in the presence of the second drug, that is, the probability that it becomes collaterally resistant (**Figure 4B**). We find that our a priori ranking corresponds perfectly to the ranking according to this probability, evidenced by the consistently higher collateral resistance risk for JDFEs with higher $c$ (**Figure 4B**). Interestingly, the top ranked JDFE does not have the lowest expected pleiotropic effect $r_2$. Nevertheless, the fact that the pleiotropic variance statistic $D_{22}$ for this JDFE is small ensures that the risk of collateral resistance evolution is the lowest. This 1–1 rank correlation holds more broadly, for all 125 Gaussian JDFEs and all population genetic parameters considered in the previous section (**Figure 4C**) as well as for the empirical TEM β-lactamase JDFEs (**Figure 4—figure supplement 1**). Overall, we find that we can use the collateral risk statistic $c$ to robustly rank drug pairs according to the risk of collateral resistance evolution, irrespective of the population genetic parameters.

## Measuring JDFEs

So far, we assumed that the parameters of the JDFE on which the population evolves are known. In reality, they have to be estimated from data, which opens up at least two practically important questions. The first question is experimental. From what types of data can JDFEs be in principle estimated and how good are different types of data for this purpose? We can imagine, for example, that some properties of JDFEs can be estimated from genome sequencing data (**Jerison et al., 2020**) or from temporally resolved fitness trajectories (**Bakerlee et al., 2021**). Here, we focus on the most direct way of estimating JDFE parameters, from the measurements of the home and non-home fitness effects of individual mutations. The experimental challenge with this approach is to sample those mutations that

will most likely contribute to adaptation in the home environment (see 'Discussion' for an extended discussion of this problem). Below, we propose two potential strategies for such sampling: the Luria-Delbrück (LD) method and the barcode lineage tracking (BLT) method. The second question is statistical: how many mutants need to be sampled to reliably rank drug pairs according to the risk of collateral resistance? We evaluate both proposed methods with respect to this property.

The idea behind the LD method is to expose the population to a given drug at a concentration above the minimum inhibitory concentration (MIC), so that only resistant mutants survive (**Pinheiro et al., 2021**). This selection is usually done on agar plates, so that individual resistant mutants form colonies and can be isolated. The LD method is relatively easy to implement experimentally, but it is expected to work only if the drug concentration is high enough to kill almost all non-resistant cells. In reality, resistant mutants may be selected at concentrations much lower than MIC (**Andersson and Hughes, 2014**). Furthermore, mutants selected at different drug concentrations may be genetically and functionally distinct (**Lindsey et al., 2013**; **Pinheiro et al., 2021**) and therefore may have statistically different pleiotropic profiles. As a result, mutants sampled with the LD method may not be most relevant for predicting collateral evolution at low drug concentrations, and other sampling methods may be required for isolating weakly beneficial mutations.

Isolating individual weakly beneficial mutations is more difficult because by the time a mutant reaches a detectable frequency in the population it has accumulated multiple additional driver and passenger mutations (**Lang et al., 2013**; **Nguyen Ba et al., 2019**), all of which can potentially have collateral effects. One way to isolate many single beneficial mutations from experimental populations is by using the recently developed barcode lineage tracking (BLT) method (**Levy et al., 2015**; **Venkataram et al., 2016**). In a BLT experiment, each cell is initially tagged with a unique DNA barcode. As long as there is no recombination or other DNA exchange, any new mutation is permanently linked to one barcode. A new adaptive mutation causes the frequency of the linked barcode to grow, which can be detected by sequencing. By sampling many random mutants and genotyping them at the barcode locus, one can identify mutants from adapted lineages even if they are rare (**Venkataram et al., 2016**). As a result, BLT allows one to sample mutants soon after they acquire their first driver mutation, before acquiring secondary mutations.

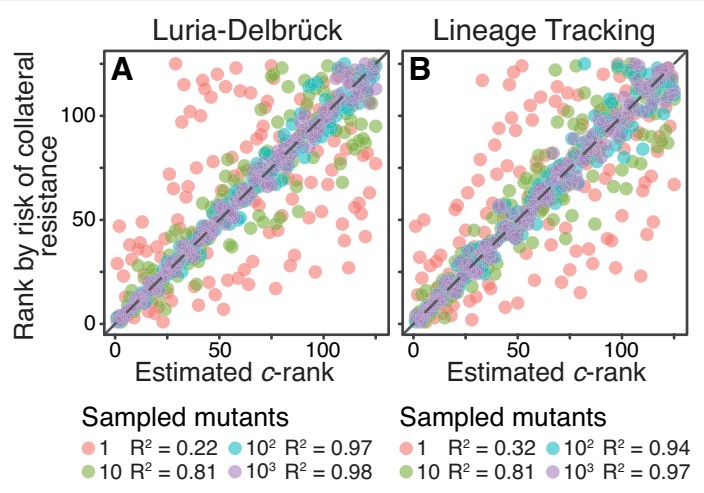

**Figure 5.** Sampling effects on the ranking of drug pairs. Both panels show correlations between the a priori estimated $c$-rank ($x$-axis) of the 125 Gaussian JDFEs and their a posteriori rank ($y$-axis) based on the risk of collateral resistance observed in simulations (same data as the $y$-axis in **Figure 4C** for $NU_b = 0.46$). (**A**) The $c$ statistic is estimated using the Luria-Delbrück method (see text for details). Cutoff for sampling mutations is $0.5\sigma$, where σ is the standard deviation of the JDFE in the home environment. See **Figure 5—figure supplement 1** for other cutoff values. (**B**) The $c$ statistic is estimated using the barcode lineage tracking method with $N = 10^6$ and $U = 10^{-4}$ (see text and Materials and methods for details). $P < 10^{-6}$ for all regressions.

The online version of this article includes the following figure supplement(s) for figure 5:

**Figure supplement 1.** Same as **Figure 5A**, but with different thresholds for sampling.

To evaluate the quality of sampling based on the LD and BLT methods, we consider the following hypothetical experimental setup. $K$ beneficial mutants are sampled from each home environment (with either one of the methods), and their home and non-home fitness $(X_i, Y_i)$ are measured for each mutant $i = 1, \ldots, K$. Since we are ultimately interested in ranking drug pairs by their risk of collateral resistance, we estimate the collateral risk statistic $\hat{c}$ from these fitness data for each drug pair and use $\hat{c}$ to rank them (see Materials and methods for details). We compare such a priori ranking of 125 hypothetical drug pairs with Gaussian JDFEs used in previous sections with their a posteriori ranking based on the risk of collateral resistance observed in simulations.

To model the LD sampling method on a given JDFE, we randomly sample $K$ mutants whose home fitness exceeds a certain cutoff. To model a BLT experiment, we simulate evolution in the home environment and randomly sample $K$ beneficial mutants segregating at generation 250 (see Materials and methods for details). We find that the $\hat{c}$-ranking estimated with either LD or BLT methods captures the a posteriori ranking surprisingly well, even when the number of sampled mutants is as low as 10 per drug pair (*Figure 5*). Given that the JDFEs with adjacent ranks differ in $c$ by a median of only 0.65%, the strong correlations shown in *Figure 5* suggest that even very similar JDFEs can be differentiated with moderate sample sizes. As expected, this correlation is further improved upon increased sampling, and it is insensitive to the specific home fitness threshold that we use in the LD method (*Figure 5— figure supplement 1*). We conclude that estimating JDFE parameters is in principle feasible with a modest experimental effort, at least for the purpose of ranking drug pairs.

## Discussion

We have shown that many resistance mutations against multiple drugs in *E. coli* exhibit a diversity of collateral effects. If this is true more generally, it implies that there is an unavoidable uncertainty in whether any given population would evolve collateral resistance or sensitivity, which could at least in part explain previously observed inconsistencies among experiments. We quantified the diversity of pleiotropic effects of mutations with a joint distribution of fitness effects (JDFE) and developed a population genetic theory for predicting the expected collateral outcomes of evolution and the uncertainty around these expectations. In the successional mutations regime, our theory shows that the average rate at which fitness in the non-home environment is gained or lost during adaptation to the home environment is determined by the pleiotropy statistic $r_2$ given by *Equation 8*. How strongly the non-home fitness in any individual population deviates from this ensemble average is determined by the pleiotropy variance statistic $D_{22}$ given by *Equation 11*. Importantly, $r_2$ and $D_{22}$ are properties of the JDFE alone, that is, they do not depend on the parameters of any specific population. In the concurrent mutations regime, the expected rate of non-home fitness gain or loss and the associated variance are reasonably well predicted by the adjusted pleiotropy statistics $r_2^*$ and $D_{22}^*$. Unlike $r_2$ and $D_{22}$, the adjusted statistics depend on the population size $N$ and the rate of beneficial mutations $U_b$.

To quantitatively predict the probability of evolution of collateral drug resistance in practice would require the knowledge of both the JDFE for the focal bacterial or cancer-cell population in the presence of the specific pair of drugs and its in vivo population genetic parameters. Since estimating the latter parameters is very difficult, it appears unlikely that we would be able to quantitatively predict the dynamics of collateral effects, even if JDFEs were known. A more realistic application of our theory is that it allows us to rank drug pairs according to the risk of collateral resistance even when the population genetic parameters are unknown. Such robust ranking can be computed based on the collateral risk statistic $c = r_2/\sqrt{D_{22}}$, a property of the JDFE but not of the evolving population. Drug pairs with positive values of $c$ have a higher chance of eliciting collateral resistance than collateral sensitivity and should be avoided; drug pairs with more negative values of $c$ have a lower risk of collateral resistance evolution than those with less negative values.

We have validated our theory in silico, but how well it would work in vivo (in the clinic) or even in vitro (in the lab) is as of yet unclear. A direct way to validate the theory empirically would be to estimate JDFEs for a model organism, such as *E. coli*, in a number of drug pairs, rank these pairs according to our collateral rank statistic and then test this ranking by evolving replicate populations and measuring the empirical distributions of collateral resistance/sensitivity outcomes. To the best of our knowledge, the antibiotic resistance JDFEs among genome-wide mutations have not yet been measured. One could in principle use existing gene knock-out data, such as those obtained by *Chevereau et al., 2015* (*Figure 1*), or the data from deep mutational scanning experiments, such as those obtained by *Stiffler*

*et al., 2015* (*Figure 1—figure supplement 1*), to estimate JDFEs. However, these experiments estimate fitness only for certain subsets of mutations (gene knock-outs or point mutations within a single gene, respectively). Since resistance may arise via other types of mutations (*Nichol et al., 2019*), these data would give us at best an incomplete picture of actual JDFEs. Our results suggest that JDFEs can be reasonably well estimated by sampling resistance mutants at drug concentrations above MIC or by employing the barcode lineage tracking method.

Another obstacle is that, even though many researchers have experimentally evolved various microbes in the presence of drugs, most experiments have maintained too few replicate populations to accurately measure the variation in collateral outcomes of evolution. The study by *Nichol et al., 2019*, with 60 replicates, is a notable exception. In short, a rigorous test of our theory requires new data on the shapes of whole-genome JDFEs as well as higher throughput evolution experiments.

What the most effective ways of measuring JDFEs are and whether it will be possible to measure JDFE in vivo are open questions. We speculate that the answers will depend on the shapes of the empirical JDFEs because some shapes may be more difficult to estimate than others. For example, if empirical JDFEs resemble multivariate Gaussian distributions, then we can learn all relevant parameters of such JDFE by sampling a handful of random mutants and measuring their fitness in relevant environments. One can also imagine more complex JDFEs where mutations beneficial in the home environment have a dramatically different distribution of non-home fitness effects than mutations that are deleterious or neutral in the home environment. In this case, very large samples of random mutations would be necessary to correctly predict the pleiotropic outcomes of evolution, so that methods that preferentially sample beneficial mutations may be required. We have considered two such methods, which are experimentally feasible. We have shown that both of them perform extremely well on Gaussian JDFEs in the sense that as few as 10 mutants per drug pair are sufficient to produce largely correct ranking of hypothetical drug pairs. However, it may be difficult to apply these methods in vivo, in which case JDFEs may have to be estimated in the lab, with selection pressures reproducing those in vivo as accurately as possible.

Our model relies on two important simplifications. It describes the evolution of an asexual population where all resistance alleles arise from de novo mutations. In reality, some resistance alleles in bacteria may be transferred horizontally (*Sun et al., 2019*). Understanding collateral resistance evolution in the presence of horizontal gene transfer events would require incorporating JDFE into other models of evolutionary dynamics (e.g. *Neher et al., 2010*). Another major simplification is in the assumption that the JDFE stays constant as the population adapts. In reality the JDFE will change over time because of the depletion of the pool of adaptive mutations and because of epistasis (*Good et al., 2017*; *Venkataram et al., 2020*). How JDFEs vary among genetic backgrounds is currently unknown. In Appendix 1, we have shown that our main results hold at least in the presence of a simple form of 'global' epistasis. Empirically measuring how JDFEs vary across genotypes and theoretically understanding how such variation affects the evolution of pleiotropic outcomes are important open questions.

While we were primarily motivated by the problem of evolution of collateral drug resistance and sensitivity, our theory is applicable more broadly. The shape of JDFE must play a crucial role in determining whether the population evolves toward a generalist or diversifies into multiple specialist ecotypes. Previous literature has viewed this question primarily through the lens of two alternative hypotheses: antagonistic pleiotropy and mutation accumulation (*Visher and Boots, 2020*). Antagonistic pleiotropy in its strictest sense means that the population is at the Pareto front with respect to the home and non-home fitness, such that any mutation beneficial in the home environment reduces the fitness in the non-home environment (*Li et al., 2019*). The shape of the Pareto front then determines whether selection would favor specialists or generalists (*Levins, 1968*; *Visher and Boots, 2020*). Alternatively, a population can evolve to become a home-environment specialist even in the absence of trade-offs, simply by accumulating mutations that are neutral in the home environment but deleterious in the non-home environment (*Kawecki, 1994*). More recently, it has been recognized that antagonistic pleiotropy and mutation accumulation are not discrete alternatives but rather extremes of a continuum of models (*Bono et al., 2020*; *Jerison et al., 2014*; *Jerison et al., 2020*). The JDFE provides a mathematical way to describe this continuum. For example, strict antagonistic pleiotropy can be modeled with a JDFE with zero probability weight in the first quadrant and a bulk of probability in the fourth quadrant. A mutation accumulation scenario can be modeled with

a '+'-like JDFE where all mutations beneficial in the home environment are neutral in the non-home environment (i.e. concentrated on the $x$-axis) and all or most mutations neutral in the home environment (i.e. those on the $y$-axis) are deleterious in the non-home environment. Our theory shows that in fact all JDFEs with negative $r_2$ lead to loss of fitness in the non-home environment and therefore can potentially promote specialization. While our theory provides this insight, further work is needed to understand how JDFEs govern adaptation to variable environments. This future theoretical work, together with empirical inquiries into the shapes of JDFEs, will not only advance our ability to predict evolution in practical situations, such as drug resistance, but it will also help us better understand the origins of ecological diversity.

## Materials and methods
### Analysis of knock-out and deep mutational scanning data
#### Knock-out data
*Chevereau et al., 2015* provide growth rate estimates for 3883 gene knock-out mutants of *E. coli* in the presence of six antibiotics. Our goal is to identify those knock-out mutations that provide resistance against one drug and are also collaterally resistant or collaterally sensitive to another drug. However, it is unclear from these original data alone which mutations have statistically significant beneficial and deleterious effects because no measurement noise estimates are provided. To address this problem, we obtained replicate wild-type growth rate measurements in the presence of antibiotics from Guillaume Chevereau and Tobias Bollenbach (available at https://github.com/ardellsarah/JDFE-project; copy archived at swh:1:rev:e91f2940681269511c6bb9fd4560ccd4a7c4d641, *Ardell, 2022*). In this additional data set, the wild-type *E. coli* strain is measured on average 476 times in the presence of each drug. We estimate the wild-type growth rate $r_{\mathrm{WT}}$ as the mean of these measurements, and we obtain the selection coefficient for all knock-out mutants as $s_i = r_i - r_{\mathrm{WT}}$. We also obtain the noise distribution $P_{\mathrm{noise}}(s)$ from the replicate wildtype measurements (shown in red in the diagonal panels in *Figure 1*). Modeling $P_{\mathrm{noise}}(s)$ as normal distributions, we obtain the $P$-values for each mutation in the presence of each antibiotic.

We then call any knock-out mutant as resistant against a given drug if its selection coefficient in the presence of that drug exceeds a critical value $s_\alpha^+ > 0$. We choose $s_\alpha^+$ using the Benjamini-Hochberg procedure (*Benjamini and Hochberg, 1995*) so that the false discovery rate (FDR) among the identified resistant mutants is $\alpha \approx 0.25$. We could not find an $s_\alpha^+$ for $\alpha \lesssim 0.25$ for trimethoprim (TMP) and chloramphenicol (CHL), that is, there were not enough knock-out mutations with positive selection coefficients to reliably distinguish them from measurement errors.

We apply the same procedure to identify mutations that are collaterally resistant and collaterally sensitive against a second drug among all mutations that are resistant against the first drug, except we aim for FDR $\lesssim 0.10$.

#### Deep mutational scanning data
*Stiffler et al., 2015* provide estimates of relative fitness for 4997 point mutations in the TEM-1 β-lactamase gene in the presence of cefotaxime (CTX) and four concentrations of ampicillin (AMP). We estimate the selection coefficients from the reported relative fitness values by changing the logarithm from $log_{10}$ to natural and dividing it by six, the estimated number of generations that occurred during the 2-hr experiment. The latter is based on the fact that *Stiffler et al., 2015* chose AMP concentrations which did not significantly alter the *E. coli* doubling time, which we assumed to be 20 minutes. We used the same number of generations for CTX.

*Stiffler et al., 2015* report two replicate measurements per mutant in each concentration of AMP and one measurement per mutant in the presence of CTX. We consider CTX as the home environment and call all mutations with positive measured fitness effects as resistant against CTX. For each such mutation, we use two replicate measurements in each concentration of AMP to estimate its mean fitness effect and the 90% confidence interval around the mean, based on the normal distribution. We call any CTX resistant mutation with the entire confidence interval above (below) zero as collaterally resistant (sensitive) against AMP at that concentration. All remaining CTX resistant mutations are called collaterally neutral.

## Theory

### Successional mutations regime

We assume that an asexual population evolves according the Wright-Fisher model in the strong selection weak mutation (SSWM) limit (*Orr, 2000*; *Kryazhimskiy et al., 2009*; *Good and Desai, 2015*), also known as the 'successional mutations' regime (*Desai and Fisher, 2007*). In this regime, the population remains monomorphic until the arrival of a new mutation that is destined to fix. The waiting time for such new mutation is assumed to be much longer than the time it takes for the mutation to fix, that is, fixation happens almost instantaneously on this time scale, after which point the population is again monomorphic. If the per genome per generation rate of beneficial mutations is $U_b$, their typical effect is $s$ and the population size is $N$, the SSWM approximation holds when $NU_b \ll 1/\ln(Ns)$ (*Desai and Fisher, 2007*).

We describe our population by a two-dimensional vector of random variables $(X_t, Y_t)$, where $X_t$ and $Y_t$ are the population's fitness (growth rate or the Malthusian parameter) in the home and non-home environments at generation $t$, respectively. We assume that the fitness vector of the population at the initial time point is known and is $(x_0, y_0)$. We are interested in characterizing the joint probability density $p(x, y, t) \, dx \, dy = \Pr\{X_t \in (x, x + dx), Y_t \in (y, y + dy)\}$.

We assume that all genotypes have the same JDFE $\Phi(\Delta x, \Delta y)$, that is, there is no epistasis. In the exponential growth model, the selection coefficient of a mutation is the difference between the mutant and the ancestor growth rates in the home environment, that is, $\Delta x$. The probability of fixation of the mutant is given by Kimura's formula, which we approximate by $2\Delta x$ for $\Delta x > 0$ and zero otherwise (*Crow and Kimura, 1972*).

If the total rate of mutations (per genome per generation) is $U$, the rate of mutations beneficial in the home environment is given by $U_b = U f_b$ where $f_b = \int_{-\infty}^{\infty} d\eta \int_0^{\infty} d\xi \, \Phi(\xi, \eta)$ is the fraction of mutations beneficial in the home environment. Once such a mutation arises, its selection coefficients in the home and non-home environments are drawn from the JDFE of mutations beneficial in the home environment $\Phi_b(\Delta x, \Delta y) = \Phi(\Delta x, \Delta y)/f_b$. Then, in the SSWM limit, our population is described by a two-dimensional continuous-time continuous-space Markov chain with the transition rate from state $(x, y)$ to state $(x', y')$ given by

$$2\mathrm{NU}_b \, Q(x', y'|x, y) = \begin{cases} 2NU_b \, (x' - x) \, \Phi_b \, (x' - x, y' - y) & \text{if } x' > x, \\ 0 & \text{otherwise.} \end{cases} \tag{3}$$

The probability distribution $p(x, y, t)$ satisfies the integro-differential forward Kolmogorov equation (*Van Kampen, 1992*)

$$\frac{1}{NU_b} \frac{\partial p}{\partial t}(x, y, t) = 2 \int_{-\infty}^{\infty} d\eta \int_{-\infty}^{\infty} d\xi \Big( p(\xi, \eta, t) \, Q(x, y|\xi, \eta) - p(x, y, t) \, Q(\xi, \eta|x, y) \Big) \tag{4}$$

with the initial condition

$$p(x, y, 0) = \delta(x - x_0) \, \delta(y - y_0). \tag{5}$$

When beneficial mutations with large effects are sufficiently rare, *Equation 4* can be approximated by the Fokker-Planck equation (*Van Kampen, 1992*)

$$\frac{1}{NU_b} \frac{\partial p}{\partial t} = -r_1 \frac{\partial p}{\partial x} - r_2 \frac{\partial p}{\partial y} + \frac{D_{11}}{2} \frac{\partial^2 p}{\partial x^2} + D_{12} \frac{\partial^2 p}{\partial x \partial y} + \frac{D_{22}}{2} \frac{\partial^2 p}{\partial y^2}, \tag{6}$$

where

$$r_1 = 2 \int_{-\infty}^{\infty} d\eta \int_0^{\infty} d\xi \, \xi^2 \, \Phi_b(\xi, \eta), \tag{7}$$

$$r_2 = 2 \int_{-\infty}^{\infty} d\eta \int_0^{\infty} d\xi \, \eta \, \xi \, \Phi_b(\xi, \eta) \tag{8}$$

are the expected fitness effects in the home and non-home environments for a mutation fixed in the home environment, and

$$D_{11} = 2 \int_{-\infty}^{\infty} d\eta \int_0^{\infty} d\xi \, \xi^3 \, \Phi_b(\xi, \eta), \tag{9}$$

$$D_{12} = 2 \int_{-\infty}^{\infty} d\eta \int_0^{\infty} d\xi \, \eta \, \xi^2 \, \Phi_b(\xi, \eta), \tag{10}$$

$$D_{22} = 2 \int_{-\infty}^{\infty} d\eta \int_0^{\infty} d\xi \, \eta^2 \, \xi \, \Phi_b(\xi, \eta) \tag{11}$$

are the second moments of the distribution of the fitness effects of mutations fixed in the home environment. The solution to *Equation 6* with the initial condition given by *Equation 5* is a multi-variate normal distribution with the mean vector $\boldsymbol{m}(t)$ and the variance-covariance matrix $\sigma^2(t)$ given by *Equation 1* and *Equation 2*.

### Concurrent mutations regime

The theory we developed so far for the successional mutations regime breaks down in the concurrent mutations regime, that is, when multiple adaptive mutations segregate in the population simultaneously (*Desai and Fisher, 2007*). The main effect of competition between segregating adaptive lineages is that many new beneficial mutations arise in relatively low-fitness genetic backgrounds and have almost no chance of surviving competition (*Desai and Fisher, 2007*; *Schiffels et al., 2011*; *Good et al., 2012*). As a result, the fixation probability of a beneficial mutation with selective effect $\Delta x$ in the home environment is no longer $2\Delta x$. Instead, beneficial mutations that provide fitness benefits below a certain threshold $x_c$ behave as if they are effectively neutral (i.e. their fixation probability is close to zero), and most adaptation is driven by mutations with benefits above $x_c$, where $x_c$ depends on the population genetic parameters $N$ and $U_b$ as well as the shape of the distribution of fitness effects of beneficial mutations. *Good et al., 2012* derived equations that allow us to calculate the effective fixation probability $\pi^*(\Delta x; N, U_b)$ of a beneficial mutation with the fitness benefit $\Delta x$ in the home environment in the concurrent mutation regime. Thus, to predict the average rate of non-home fitness change, we replace the SSWM fixation probability $2\xi$ in *Equation 8* with $\pi^*(\xi; N, U_b)$ and obtain the adjusted expected pleiotropic effect. We similarly obtain the adjusted pleiotropic variance statistic

$$D_{22}^*(N, U_b) = \int_{-\infty}^{\infty} d\eta \int_0^{\infty} d\xi \, \eta^2 \, \pi^*(\xi; N, U_b) \, \Phi_b(\xi, \eta), \tag{13}$$

although as discussed in Section 'The population genetics of pleiotropy', we do not expect $D_{22}^*$ to capture all of the variation in non-home fitness trajectories.

To calculate $\pi^*(\Delta x; N, U_b)$ for the Gaussian JDFEs shown in *Figure 2*, we first substitute Equation (20) in *Good et al., 2012* with $\beta = 2$ into Equation 18, 19 in *Good et al., 2012* and then numerically solve these equations for $x_c$ and $v$ using the FindRoot numerical method in Mathematica. Note that all our Gaussian JDFEs share the same mean and variance in the home environment, so we need to solve these equations only once for each pair of $N$ and $U_b$ values. We then substitute the obtained values of $x_c$ and $v$ into Equation (4) (9) in *Good et al., 2012* and calculate $\pi^*$ by a numerical integration of Equation (6) in *Good et al., 2012* in R (available at https://github.com/ardellsarah/JDFE-project).

### Ranking of drug pairs

According to *Equation 1* and *Equation 2*, both the expected non-home fitness and its variance change linearly with time, so that at time $t$ the mean is $Z = c\sqrt{NU_b t}$ standard deviations above $y_0$ (if $r_2 > 0$) or below $y_0$ (if $r_2 < 0$), where $c = r_2/\sqrt{D_{22}}$. In other words, if $r_2 > 0$, the bulk of the non-home fitness distribution eventually shifts above $y_0$, and if $r_2 < 0$, it shifts below $y_0$. All else being equal, a larger value of $|c|$ implies faster rate of this shift.

The interpretation of these observations in terms of collateral resistance/sensitivity is that adaptation in the presence of the first drug will eventually lead to collateral resistance against the second drug if $r_2 > 0$ and to collateral sensitivity if $r_2 < 0$. Furthermore, all else being equal, collateral sensitivity evolves faster and the chance of evolving collateral resistance is smaller for drug pairs with more negative $c$ (i.e. larger $|c|$). Thus, we use $c$ to order drug pairs from the most preferred (those with the most negative values of $c$) to least preferred (those with least negative or positive values of $c$).

## Generation of JDFEs

### Gaussian JDFEs

The JDFEs in *Figure 2* have the following parameters. Mean in the home environment: -0.05. Standard deviation in both home and non-home environments: 0.1. Means in the non-home environment: 0.08, 0.145, 0, -0.145, -0.08 in panels A through E, respectively.

The JDFEs in *Figure 3* have the following parameters. Mean and standard deviation in the home environment: -0.001 and 0.001, respectively. The non-home mean varies between 0.0001 and 0.01. The non-home standard deviation varies between 0.0001 and 0.01. The correlation between home and non-home fitness varies between -0.9 and 0.9, for a total of 125 JDFEs. All parameter values and the resulting pleiotropy statistics for these JDFEs are given in the *Figure 3—source data 1*.

### JDFEs with equal probabilities of pleiotropically beneficial and deleterious mutations

All JDFEs in *Figure 2—figure supplement 1* are mixtures of two two-dimensional uncorrelated Gaussian distributions, which have the following parameters. Mean in the home environment: 0.4. Standard deviation in both home and non-home environments: 0.1. Means in the non-home environment: 0.1 and -0.1 in panel A, 0.5 and -0.5 in panel B, 0.17 and -0.5 in panel C, and 0.5 and -0.17 in panel D.

## Simulations

We carried out two types of simulations, SSWM model simulations and full Wright-Fisher model simulations.

### Strong selection weak mutation

The SSWM simulations were carried out using the Gillespie algorithm (*Gillespie, 1976*), as follows. We initiate the populations with home and non-home fitness values $x_0 = 0$ and $y_0 = 0$. At each iteration, we draw the waiting time until the appearance of the next beneficial mutation from the exponential distribution with the rate parameter $NU_b$ and advance the time by this amount. Then, we draw the selection coefficients $\Delta x$ and $\Delta y$ of this mutation in the home- and non-home environment, respectively, from the JDFE (a multivariate normal distribution). With probability $2\Delta x$, the mutation fixes in the population. If it does, the fitness of the population is updated accordingly.

### Wright-Fisher model

We simulate evolution in the home environment according to the Wright-Fisher model with population size $N$ as follows. We initiate the whole population with a single genotype with fitness $x_0 = 0$ and $y_0 = 0$ in the home and non-home environments. Suppose that at generation $t$, there are $K(t)$ genotypes, such that genotype has home- and non-home fitness $X_i$ and $Y_i$, respectively, and it is present at frequency $f_i(t) > 0$ in the population. We generate the genotype frequencies at generation $t + 1$ in three steps. In the reproduction step, we draw random numbers $B'_i(t + 1)$, $i = 1, ..., K(t)$ from the multinomial distribution with the number of trials $N$ and success probabilities $p_i(t) = f_i(t) + f_i(t) \left( X_i(t) - \overline{X}(t) \right)$, where $\overline{X}(t) = \sum_{i=1}^{K(t)} X_i(t) f_i(t)$ is the mean fitness of the population in the home environment at generation $t$. In the mutation step, we draw a random number $M$ of new mutants from the Poisson distribution with parameter $NU$, where $U$ is the total per individual per generation mutation rate. We randomly determine the 'parent' genotypes in which each mutation occurs and turn the appropriate numbers of parent individuals into new mutants. We assume that each new mutation creates a new genotype and has fitness effects $\Delta x$ and $\Delta y$ in the home and non-home environments. $\Delta x$ and $\Delta y$ are drawn randomly from the JDFE $\Phi(\Delta x, \Delta y)$. We obtain each mutants fitness by adding these values to the parent genotype's home and non-home fitness values. In the final step, all genotypes that are represented by zero individuals are removed and we are left with $K(t + 1)$ genotypes with $B_i(t + 1) > 0$ individuals, $i = 1, \dots, K(t + 1)$. Then we set $f_i(t + 1) = B_i(t + 1)/N$.

## Sampling beneficial mutants from JDFEs and estimating the collateral risk statistic

We model the LD sampling method by randomly drawing mutants from the JDFE until the desired number $K$ of mutants whose home fitness exceeds the focal threshold are sampled. We estimate the $c$ statistic from the pairs of home and non-home fitness effects $X_i$ and $Y_i$ of these $i = 1, \ldots, K$ sampled mutants. To do so, we first estimate $r_2$ and $D_{22}$ as $\hat{r}_2 = 1/K \sum_{i=1}^{K} X_i Y_i$ and $\hat{D}_{22} = 1/K \sum_{i=1}^{K} X_i Y_i^2$. We then calculate $\hat{c} = \hat{r}_2 / \sqrt{\hat{D}_{22}}$.

For the BLT sampling method, we simulate the Wright-Fisher model as described above for $N = 10^6$ and $U = 10^{-4}$ for 250 generations. At generation 250, we randomly sample existing beneficial mutants proportional to their frequency in the population without replacement (i.e. the same beneficial mutation is sampled at most once). Sampling more than $\sim 50$ distinct beneficial mutants from a single population becomes difficult because there may simply be not enough such mutants or some of them may be at very low frequencies. Therefore, if the desired number of mutants to sample exceeds 50, we run multiple replicate simulations and sample a maximum of 100 distinct beneficial mutants per replicate until the desired number of mutants is reached. We then estimate the $c$ statistics as with the LD method.

## Acknowledgements

We thank Shea Summers and Flora Tang for help and input at the initial stages of the project and Kryazhimskiy and Meyer labs for feedback. We thank Tobias Bollenbach and Guillaume Chevereau for providing wildtype growth rate measurement data. This work was supported by the BWF Career Award at Scientific Interface (Grant 1010719.01), the Alfred P Sloan Foundation (Grant FG-2017–9227), the Hellman Foundation and NIH (Grants 1R01GM137112 and 1T32GM133351-01).

## Additional information

### Funding

| Funder | Grant reference number | Author |
|---|---|---|
| Burroughs Wellcome Fund | 1010719.01 | Sergey Kryazhimskiy |
| Alfred P. Sloan Foundation | FG-2017-9227 | Sergey Kryazhimskiy |
| Hellman Foundation | | Sergey Kryazhimskiy |
| National Institutes of Health | 1R01GM137112 | Sergey Kryazhimskiy |
| National Institutes of Health | 1T32GM133351-01 | Sarah M Ardell |

The funders had no role in study design, data collection and interpretation, or the decision to submit the work for publication.

### Author contributions

Sarah M Ardell, Conceptualization, Data curation, Formal analysis, Investigation, Methodology, Software, Visualization, Writing - original draft, Writing - review and editing; Sergey Kryazhimskiy, Conceptualization, Formal analysis, Funding acquisition, Investigation, Methodology, Project administration, Supervision, Visualization, Writing - original draft, Writing - review and editing

### Author ORCIDs

Sarah M Ardell http://orcid.org/0000-0001-6665-2739
Sergey Kryazhimskiy http://orcid.org/0000-0001-9128-8705

### Decision letter and Author response

Decision letter https://doi.org/10.7554/eLife.73250.sa1
Author response https://doi.org/10.7554/eLife.73250.sa2

# Additional files

## Supplementary files
• Transparent reporting form

## Data availability
All code is available on GitHub (https://github.com/ardellsarah/JDFE-project; copy archived at swh:1:rev:e91f2940681269511c6bb9fd4560ccd4a7c4d641). All data are available as Source Data files, included with the manuscript.

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

## Appendix 1

### JDFE with global epistasis

Results in the main text were derived under the assumption that all genotypes have the same JDFE, i.e., in the absence of epistasis. In reality, JDFEs probably vary from one genotype to another, but how they vary is not yet well characterized. Recent studies have found that the fitness effects of many mutations available to a genotype in a given environment depend primarily on the fitness of that genotype in that environment (*Khan et al., 2011*; *Chou et al., 2011*; *Wiser et al., 2013*; *Kryazhimskiy et al., 2014*; *Johnson et al., 2019*; *Wang et al., 2016*; *Aggeli et al., 2021*; *Lukačišinová et al., 2020*). This dependence is sometimes referred to as global or fitness-dependent epistasis (*Kryazhimskiy et al., 2009*; *Kryazhimskiy et al., 2014*; *Reddy and Desai, 2021*; *Husain and Murugan, 2020*). Here, we ask whether our main results would hold if the pathogen population evolves on a JDFE with global epistasis.

Global epistasis can be modeled in our framework by assuming that the JDFE $\Phi_g$ of genotype $g$ depends only the fitness of this genotype in the home and non-home environments, $x(g)$, $y(g)$, i.e. $\Phi_g\left(\Delta x, \Delta y\right) = \Phi_{x(g),y(g)}\left(\Delta x, \Delta y\right)$, which is a two-dimensional extension of the model considered by *Kryazhimskiy et al., 2009*. Thus, in the SSWM regime, the population can still be fully described by its current pair of fitness values in the home and non-home environments $(X_t, Y_t)$. The dynamics of the probability density $p(x, y, t)$ are governed by the same Kolmogorov equation as in the non-epistatic case, which can still be approximated by a diffusion equation (*Equation 6*). However, while in the non-epistatic case the drift and diffusion coefficients of this equation, $r_1$, $r_2$, $D_{11}$, $D_{12}$ and $D_{22}$ are constants, in the presence of global epistasis, they become functions of $x$ and $y$. Although this equation cannot be solved analytically in the general case, it can be solved numerically, provided that the functions $r_1(x, y)$, $r_2(x, y)$, $D_{11}(x, y)$, $D_{12}(x, y)$ and $D_{22}(x, y)$ are known. Thus, in principle, our theory can predict the trajectories of non-home fitness in the presence of global epistasis.

To explore the implications of global epistasis for collateral drug resistance evolution, we consider the simplest scenario where the functional form of global epistasis (i.e., how $\Phi_{x,y}$ depends on $x$ and $y$) is the same across different drugs. In this case, we would expect that the ranking of drug pairs according to the risk of collateral resistance would be the same for all genotypes. In particular, the drug pair whose risk of collateral resistance risk is the lowest for the wildtype should also be the pair with the lowest risk for the evolved genotypes.

To test this prediction, we model resistance evolution on Gaussian JDFEs whose mean vector and the correlation coefficient are fixed while the standard deviations and in the home and non-home environments decrease linearly with the fitness in the respective environment, $\sigma_h(x) = max\{0, \sigma_{h,0} - \gamma_h x\}$ and $\sigma_{nh}(y) = max\{0, \sigma_{nh,0} - \gamma_{nh}y\}$.

*Appendix 1—figure 1A* shows how one such JDFE changes along the expected evolutionary trajectory. The corresponding expected home and non-home fitness trajectories and their variance are shown in *Appendix 1—figure 1B*. *Appendix 1—figure 1C* shows how the probability (risk) of collateral resistance changes over time on four different JDFEs with global epistasis. For the ancestral strain (whose fitness we set by convention to $x = y = 0$), these four JDFEs are identical to those shown in *Figure 4A*; as the populations evolve, JDFEs change as specified above with $\gamma_h = \gamma_{nh} = 0.5$. As expected, the ranking of these epistatic JDFEs according to the risk of collateral resistance stays constant over time and can be predicted from estimates of the $c$ parameters for the ancestral strain.

To test this prediction, we model resistance evolution on Gaussian JDFEs whose mean vector and the correlation coefficient are fixed while the standard deviations and in the home and non-home environments decrease linearly with the fitness in the respective environment, $\sigma_h(x) = max\{0, \sigma_{h,0} - \gamma_h x\}$ and $\sigma_{nh}(y) = max\{0, \sigma_{nh,0} - \gamma_{nh}y\}$.*Appendix 1—figure 1A* shows how one such JDFE changes along the expected evolutionary trajectory. The corresponding expected home and non-home fitness trajectories and their variance are shown in *Appendix 1—figure 1B*. *Appendix 1—figure 1C* shows how the probability (risk) of collateral resistance changes over time on four different JDFEs with global epistasis. For the ancestral strain (whose fitness we set by convention to $x = y = 0$), these four JDFEs are identical to those shown in *Figure 4A*; as the populations evolve, JDFEs change as specified above with $\gamma_h = \gamma_{nh} = 0.5$. As expected, the ranking of these epistatic JDFEs according to the risk of

collateral resistance stays constant over time and can be predicted from estimates of the $c$ parameters for the ancestral strain.

To test this prediction, we model resistance evolution on Gaussian JDFEs whose mean vector and the correlation coefficient are fixed while the standard deviations and in the home and non-home environments decrease linearly with the fitness in the respective environment, $\sigma_h(x) = max\{0, \sigma_{h,0} - \gamma_h x\}$ and $\sigma_{nh}(y) = max\{0, \sigma_{nh,0} - \gamma_{nh} y\}$.*Appendix 1—figure 1A* shows how one such JDFE changes along the expected evolutionary trajectory. The corresponding expected home and non-home fitness trajectories and their variance are shown in *Appendix 1—figure 1B*. *Appendix 1—figure 1C* shows how the probability (risk) of collateral resistance changes over time on four different JDFEs with global epistasis. For the ancestral strain (whose fitness we set by convention to $x = y = 0$), these four JDFEs are identical to those shown in *Figure 4A*; as the populations evolve, JDFEs change as specified above with $\gamma_h = \gamma_{nh} = 0.5$. As expected, the ranking of these epistatic JDFEs according to the risk of collateral resistance stays constant over time and can be predicted from estimates of the $c$ parameters for the ancestral strain.

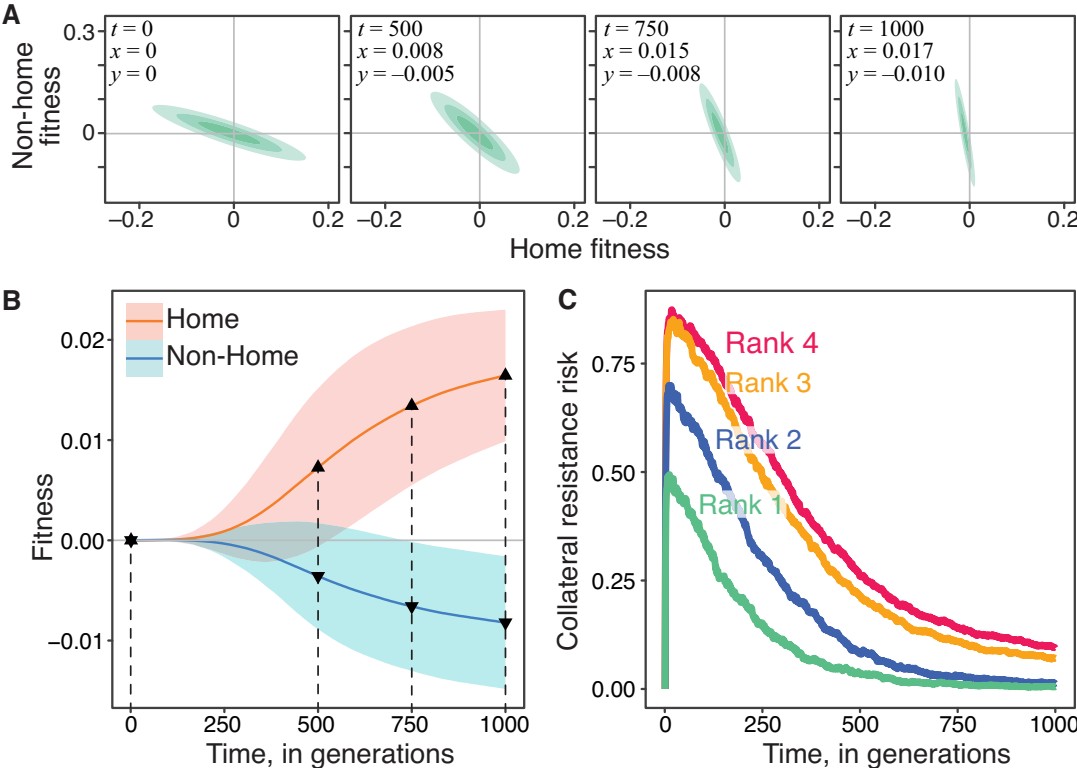

**Appendix 1—figure 1.** Evolution on JDFEs with global epistasis and the risk of collateral resistance. (**A**) Gaussian JDFE with global epistasis as it changes along the expected evolutionary trajectory shown in panel B. Parameters of the initial JDFE at $x = y = 0$ are the same as for the ank 1 JDFE in *Figure 4A*; $\gamma_h = \gamma_{nh} = 0.5$. (**B**) Home and non-home fitness trajectories for the JDFE with global epistasis shown in panel A. Thick lines show the mean, ribbons show ±1 standard deviation estimated from 500 replicate simulations. Population size $N = 10^4$, mutation rate $U = 10^{-4}$. Dashed vertical lines indicate the time points at which the JDFE snapshots in panel A are shown. (**C**) Probability of collateral resistance over time for four Gaussian JDFE with global epistasis. Parameters of the initial JDFEs at $x = y = 0$ are the same as for the four JDFE in *Figure 4A*, and $\gamma_h = \gamma_{nh} = 0.5$ for all of them. $N = 10^4$, mutation rate $U = 10^{-4}$, 1500 replicate simulation runs per JDFE. Colored numbers indicate the predicted -rank of the initial JDFEs (same as in *Figure 4A*).

