## [Editor Report]

When selecting for one particular trait, it is not uncommon for other traits to change. This is due to pleiotropic mutations that affect multiple characters. Ardell and Kryazhimskiy develop a theoretical framework to predict adaptive trajectories observed in environments other than the one selection is operating in. The effects of adaptation across environments have important implication to antibiotic treatments, where resistance evolution to one antibiotic can alter the susceptibility to other antibiotics.

---

## [Decision Letter]

**Decision letter after peer review:**

[Editors’ note: the authors submitted for reconsideration following the decision after peer review. What follows is the decision letter after the first round of review.]

Thank you for submitting your work entitled "The population genetics of pleiotropy, and the evolution of collateral resistance and sensitivity in bacteria" for consideration by *eLife*. Your article has been reviewed by 3 peer reviewers, including Richard A Neher as the Reviewing Editor and Reviewer #1, and the evaluation has been overseen by a Senior Editor. The following individual involved in review of your submission has agreed to reveal their identity: Joachim Krug (Reviewer #2).

Our decision has been reached after consultation between the reviewers. Based on these discussions and the individual reviews below, we regret to inform you that your work will not be considered further for publication in *eLife*. However, if you feel you can address the comments as outlined below, we encourage you to resubmit.

The reviewers agreed that pleiotropy of mutations and the resulting adaptive trajectories across different environment are important topics that are both of theoretical and applied interest. Your theoretical framework predicts fitness trajectories observed in environments other than the one selection is operating in (home env). These trajectories in non-home environments are calculated via integrals over the joint fitness effect distribution weighted by the fixation probability in the home environment. However, your framework assumes strong selection and weak mutation (SSWM) and deviations from this assumption seem to have strong effects. We think that these effects need to be at least partially understood. Furthermore, application to the KO library is a useful proof-of-concept, but the practical relevance of these patterns for understanding collateral sensitivity/resistance is far from obvious.

In summary, we felt that the manuscript needs to make a more substantiative theoretical advances and/or provide more robust actionable insights into drug resistance evolution to justify publication in *eLife*. We would be happy to reconsider this manuscript should you be able to make substantive progress towards these issues.

*Reviewer #1:*

Ardell and Kryazhimskiy use bacterial KO data in multiple conditions to study the structure of pleiotropy, that is the degree to which a genetic perturbation affects multiple phenotypes, and present a theoretical framework to predict and assess fitness trajectories observed in environments other than the one selection is operating in. The work is thoroughly done and has potentially interesting implications for sequential drug therapy.

The central object of their framework is the joint distribution of fitness effects of mutations in multiple environments where the distribution is over all mutations in the genome. The dynamics in the space of fitness in multiple environments is then modeled as a random walk (described by a diffusion equation) assuming that mutations sweep separated in time (SSWM). The model and the calculations necessary to arrive at the predictions are simple and transparent. The results quantitatively predict simulation results with the range of validity of SSWM. Outside this range, the model predicts the qualitative behavior, but is quantitatively wrong.

1) My main disappointment with the paper is the inability to quantitatively describe the dynamics outside the SSWM regime. I would expect that the effects of competing mutations or weak selection could be accounted for at least perturbatively. Alternative, one could determine the distribution of the effects of fixed mutations in the "home" environment in simulations and use this distribution to predict the dynamics in other environments.

2) My other substantial concern is the question whether anything can be learned about drug resistance evolution or collateral sensitivity/resistance from KO experiments. While some drug resistance evolution involves loss-of-function mutations (e.g. porin losses), it often proceeds via point mutations, up-regulation, or horizontal acquisition. Furthermore, the statistical treatment here requires many mutations to sample the joint effect distribution to give reliable answers. In clinical resistance evolution, the number of mutations observed is often quite small and their effect distributions are wide. The practical relevance of this is therefore far from clear.

3) While the similarity of this work to similar questions in quantitative genetics is discussed in the introduction, I would like to see an extended discussion whether some limits of the model at hand can be described by the quantitative genetics approach.

*Reviewer #2:*

The authors present a theoretical framework for analysing pleiotropic effects in populations evolving in different environments based on the concept of a joint distribution of fitness effects (JDFE). Simple correlation measures are derived from the JDFE that allow one to predict the evolutionary outcome in the non-home environment. Analytic theory is derived in the SSWM regime and complemented by simulations covering the regime of large mutation supply. A proof-of-concept application to collateral antibiotic resistance and sensitivity in bacteria based on a published data set for knockout strains is presented. Overall, this is an important, systematic contribution to very timely subject that is well suited for publication in *eLife*.

1. I do not quite share the authors' surprise at the outcomes shown in Figure 1. In fact, there is a simple heuristic that allows one to predict the direction of the fitness change in the non-home environment in all cases: Simply look at the y-coordinate of the tail of the JDFE corresponding to the largest beneficial effects along the x-axis.

2. Along the three rows of panels in Figure 2, there appears to be a systematic but in two cases non-monotonic variation of the slope with the mutation supply NU_b. Do the authors have a (tentative) explanation for this behavior?

*Reviewer #3:*

The goal of this manuscript--to develop predictive tools for inferring fitness trajectories in new environments--is an important goal and I appreciate the synthesis of theoretical modeling with parameter estimation from empirical mutation studies.

Reading through the manuscript, however, I found myself repeatedly wondering whether the stated application of the methods developed here doesn't constitute something of a tautology. This could be a misreading on my end, but I'll explain: the authors state that they have the central goal of predicting whether a population adapting to one environment will lose fitness in another "non-home" environment. Yet the parameter estimation they develop and propose for estimating fitness trajectories requires fitness measurements in both the home and non-home environments. If one already has fitness measurements for both home and non-home, how much more information is added by estimating the JDFE? I understand that the authors are estimating the fitness trajectories over time, with the incorporation of population genetic parameters, but again, I was unsure of how much information was added with the JDFE particularly given large discrepancies in the Wright-Fisher models and the decreasing predictive capacity with time. The bottom row of Figure 1 provided perhaps the most convincing evidence of the usefulness of the JDFE, but the unintuitive result was not adequately explored nor explained (see comment below). Also, perhaps an exploration of how the predictions could be extended to unmeasured environments is possible (as in Kinsler et al. 2020)?

Further specific conceptual comments and suggestions:

1) The authors demonstrate in Figure 1 that JDFEs even with similar shapes produce markedly different fitness trajectories. They argue that the correlation coefficient of the JDFE is not a reliable predictor of fitness trajectories in the home environment. I was struck by this counterintuitive result, and found myself searching for further explanation. Are the authors arguing that the practice of simply looking at the correlation coefficient in tradeoff studies in general is insufficient for predicting the fates of pleiotropic mutations? Either way, it would be helpful to the reader to elaborate on why and under which conditions the discrepancy with the correlation coefficient and fitness trajectories arises.

2) The modeling results throughout the manuscript reveal poor predictive capabilities in Wright-Fisher simulations. For example, the results in figure 2 show substantial discrepancy between the theoretical predictions and the results of the Wright-Fisher simulations. The authors address this only briefly stating that outside of the strong selection, weak mutation model (SSWM) the pleiotropy statistics are only "statistical predictors". But the discrepancy was systematic and wide, suggesting rather little insight from the pleiotropy statistics in sequential adaptation scenarios. I could not find discussion of this discrepancy between the SSWM and Wright-Fisher modeling predictions.

[Editors’ note: further revisions were suggested prior to acceptance, as described below.]

Thank you for resubmitting your work entitled "The Population Genetics of Collateral Resistance and Sensitivity" for further consideration by *eLife*. Your revised article has been evaluated by Meredith Schuman (Senior Editor) and a Reviewing Editor.

The manuscript has been improved considerably. The reviewers particularly appreciated the more general theoretical results. However, we would like to see a more thorough discussion of previous literature. In particular, the study by Nichol et al. is important prior work that needs to be discussed in greater depth.

In addition to a more extensive discussion of the literature, we would also like to see, if at all possible, some level of empirical validation of the results beyond the KO data presented so far. The data by Stiffler et al. and Nichol et al. characterize point mutations. Using these data beyond what is currently shown in Figure S1 could be very valuable.

*Reviewer #2:*

The manuscript has been largely restructured and rewritten, and has improved considerably. The issues that I raised in my previous report have been adequately addressed. As requested in the previous round of reviews, the theory has been generalized beyond the SSWM regime. Moreover, a second data set on resistance mutations in β-lactamase has been added, and the discussion of the practical applicability of the approach has been extended significantly. As far as I am concerned, the paper can be accepted in its present form.

*Reviewer #3:*

The authors thoroughly responded to the reviewers' comments and I found the resubmission to be both clearer and to demonstrate greater prediction accuracy in the Wright Fisher simulations. The addition of the section on estimating JDFE parameters from experimental data was a positive addition to the manuscript in that it provides a bridge for experimentalists to implement the methods developed by the authors.

That being said, as an experimentalist who could potentially implement the proposed modeling in my own work for predicting tradeoffs, I am not yet convinced of the significant advance of the proposed modeling framework for making predictions. Specifically, I found the following two points to present the most significant drawbacks to the manuscript at present:

i) I found the manuscript to lack sufficient discussion of what has been shown before in the field of modeling collateral resistances and how the present manuscript presents a clear advance in light of this work. To the first point, a brief perusal of recent literature on collateral resistance brought me to Nichol et al. 2019 Nature Comm. Ardell et al. reference the Nichol manuscript on line 37 when stating that previous work observes wide variation in collateral outcomes. But Nichol et al. did more than demonstrate variation in collateral outcomes, and instead conducted 60 parallel experimental evolution assays in one antibiotic, measured the probability of collateral resistance/susceptibility and then modeled through SSWM simulations the predicted collateral resistance outcomes for dozens of drug pairs. The present manuscript should explain how its methods/goals/results differ from those of Nichol et al.

My second point is (ii) the manuscript would be significantly strengthened if it could provide proof-of-concept validations beyond the KO work and the β-lactamase work. If I understand correctly, the authors perform the drug-ranking experiments with simulated data. I am surprised that the authors cannot find a dataset in which to validate any part of the drug-ranking predictions. This type of validation would be helpful in convincing the reader of the strength of the proposed methods. As a relevant aside, Beyond Figure S1 I couldn't find where the Β-lactamase data was used and the basic conclusion stated in the text for S1 regarding variable resistance pleiotropy is already well-established in previous work.

---

## [Author Response]

[Editors’ note: the authors resubmitted a revised version of the paper for consideration. What follows is the authors’ response to the first round of review.]

The reviewers agreed that pleiotropy of mutations and the resulting adaptive trajectories across different environment are important topics that are both of theoretical and applied interest. Your theoretical framework predicts fitness trajectories observed in environments other than the one selection is operating in (home env). These trajectories in non-home environments are calculated via integrals over the joint fitness effect distribution weighted by the fixation probability in the home environment. However, your framework assumes strong selection and weak mutation (SSWM) and deviations from this assumption seem to have strong effects. We think that these effects need to be at least partially understood. Furthermore, application to the KO library is a useful proof-of-concept, but the practical relevance of these patterns for understanding collateral sensitivity/resistance is far from obvious.In summary, we felt that the manuscript needs to make a more substantiative theoretical advances and/or provide more robust actionable insights into drug resistance evolution to justify publication in eLife. We would be happy to reconsider this manuscript should you be able to make substantive progress towards these issues.Reviewer #1:Ardell and Kryazhimskiy use bacterial KO data in multiple conditions to study the structure of pleiotropy, that is the degree to which a genetic perturbation affects multiple phenotypes, and present a theoretical framework to predict and assess fitness trajectories observed in environments other than the one selection is operating in. The work is thoroughly done and has potentially interesting implications for sequential drug therapy.The central object of their framework is the joint distribution of fitness effects of mutations in multiple environments where the distribution is over all mutations in the genome. The dynamics in the space of fitness in multiple environments is then modeled as a random walk (described by a diffusion equation) assuming that mutations sweep separated in time (SSWM). The model and the calculations necessary to arrive at the predictions are simple and transparent. The results quantitatively predict simulation results with the range of validity of SSWM. Outside this range, the model predicts the qualitative behavior, but is quantitatively wrong.1) My main disappointment with the paper is the inability to quantitatively describe the dynamics outside the SSWM regime. I would expect that the effects of competing mutations or weak selection could be accounted for at least perturbatively. Alternative, one could determine the distribution of the effects of fixed mutations in the "home" environment in simulations and use this distribution to predict the dynamics in other environments.

We agree with both Dr. Neher and reviewer 3 in their assessment of the quantitative disagreement between theory and simulations. Although the development of a full theory of evolution on a JDFE in the concurrent mutations regime goes beyond the scope of this paper, we were able to extend our main results to this regime by using the adjusted fixation probability derived by Good et al. (PNAS 2012). This simple adjustment now produces a good quantitative agreement between theory and simulations.

While developing a quantitative theory of pleiotropy is important, it is also important to keep in mind that quantitative predictions would likely be impossible in many practical situations (such as when decisions need to be made with which drugs to treat a patient) because the population genetic parameters of the treated population will not be known. In such practical situations, a ranking of drug pairs by their risk of collateral resistance evolution may be more useful than a quantitative theory, as long as the ranking is robust with respect to the population genetic parameters. We discuss this problem and our solution to it explicitly in the new section “Robust ranking of drug pairs”.

2) My other substantial concern is the question whether anything can be learned about drug resistance evolution or collateral sensitivity/resistance from KO experiments. While some drug resistance evolution involves loss-of-function mutations (e.g. porin losses), it often proceeds via point mutations, up-regulation, or horizontal acquisition. Furthermore, the statistical treatment here requires many mutations to sample the joint effect distribution to give reliable answers. In clinical resistance evolution, the number of mutations observed is often quite small and their effect distributions are wide. The practical relevance of this is therefore far from clear.

Dr. Neher raises two important concerns about the practical relevance of our work. (1) Relevance of the knock-out mutations for predicting resistance evolution. (2) Potential difficulties in estimating JDFEs from limited data for the JDFE framework to be useful. Overall, we have reorganized and augmented the manuscript to make it more practically relevant, specifically re-focusing it on the evolution of collateral resistance and sensitivity. To address the two specific issues raised by the reviewer, we did the following.

1) First, we note that Chevereau et al. (PLoS Biol, 2015) show that the answer to “the question whether anything can be learned about drug resistance evolution” from KO experiments is “yes”, at least in some cases (see Figure 4 in their paper). This important result notwithstanding, we agree with the reviewer that it is in general unclear what KO mutations alone tell us about full JDFEs. Our presentation in the original manuscript suggested a more direct applicability of koJDFEs than intended. Instead, our central thesis is that the JDFE framework is useful for thinking about collateral resistance/sensitivity evolution, not that actual JDFEs can be estimated from knock-out mutations. To make this point clear, we made the following changes.

a) We removed any discussion of the pleiotropy parameters of koJDFE. In fact, we no longer use the term koJDFE.

a) We moved the presentation of the KO data to the beginning of the Results section. We now use these data to demonstrate that resistance mutations are highly diverse in terms of their collateral effects. This fact may seem obvious in retrospect, but, to our knowledge, it has never been made explicitly in the antibiotic resistance literature. We use this argument to justify the JDFE framework for modeling collateral resistance/sensitivity.

c) To further bolster this argument, we now use a second data set, the mutational scanning data in the TEM-1 β-lactamase gene, obtained by Stiffler et al. (Cell 2015). We show that even point mutations in a single gene exhibit a great degree of variability in their collateral effects.

2) First, we want to clarify that for the JDFE framework to be useful the JDFE need not consist of many mutations. For example, a JDFE can be discrete. But as long as there are multiple classes of mutations with different pleiotropic effects, the pleiotropic outcomes of evolution will be uncertain. Understanding this uncertainty and taking it into account for designing drug treatments requires knowing the probabilities with which different classes of mutations arise, which is the JDFE.

Having said that, we agree with the reviewer that, if JDFEs cannot be reliably estimated, the utility of this framework would be greatly diminished. To address this issue, we added a new section “Measuring JDFEs” where we show how JDFEs can be estimated for practical purposes of drug treatment choice from relatively little data, at least if the JDFEs have relatively simple shapes.

3) While the similarity of this work to similar questions in quantitative genetics is discussed in the introduction, I would like to see an extended discussion whether some limits of the model at hand can be described by the quantitative genetics approach.

We have extensively revised the manuscript, including the Discussion, but we are not quite sure what limits Dr. Neher has in mind here. If he still thinks it is necessary to discuss some additional points, we would appreciate any specific suggestions.

Reviewer #2:The authors present a theoretical framework for analysing pleiotropic effects in populations evolving in different environments based on the concept of a joint distribution of fitness effects (JDFE). Simple correlation measures are derived from the JDFE that allow one to predict the evolutionary outcome in the non-home environment. Analytic theory is derived in the SSWM regime and complemented by simulations covering the regime of large mutation supply. A proof-of-concept application to collateral antibiotic resistance and sensitivity in bacteria based on a published data set for knockout strains is presented. Overall, this is an important, systematic contribution to very timely subject that is well suited for publication in eLife.1. I do not quite share the authors' surprise at the outcomes shown in Figure 1. In fact, there is a simple heuristic that allows one to predict the direction of thefitness change in the non-home environment in all cases: Simply look at the y-coordinate of the tail of the JDFE corresponding to the largest beneficial effects along the x-axis.

While Dr. Krug did not find the observations in Figure 2 (former Figure 1) surprising, Reviewer 3 did. The degree of surprise naturally varies, but it is quite certain that the majority of *eLife* readers would not have as well developed intuition about evolution as Dr. Krug. We therefore believe that this figure will help many readers to better appreciate the complexity of the problem at hand. Specifically, one important point that this figure makes is that the correlation between mutational effects—a commonly used metric of trade-offs—does not predict pleiotropic outcomes of evolution.

As a side note, if we understand the heuristic suggested by Dr. Krug correctly, it may have been an artifact of plotting. After this comment, we noticed that the tip of the outermost contour line in the original version of the figure did indeed predict the direction of collateral evolution. But of course this line is arbitrary, and how to identify “the largest beneficial effects along the x-axis” is unclear. We have now changed the spacing between the contour lines so that there are no more spurious relationships (that we can see) between these lines and the observed collateral outcomes of evolution.

2. Along the three rows of panels in Figure 2, there appears to be a systematic but in two cases non-monotonic variation of the slope with the mutation supply NU_b. Do the authors have a (tentative) explanation for this behavior?

Thank you for pointing out this issue. An investigating of this issue led us to discover an error in Equation (3). In this equation, the full JDFE had to be replaced with the JDFE of mutations beneficial in the home environment, which we now did. After correcting this error, we find that the SSWM predictions always overestimate the actual rate of fitness change in the home and in the non-home environments, as they should (see Figure S3). As a consequence, all data points in the first quarter of the plots in Figure S3 (these data were presented in Figure 2 in the previous version of the manuscript) are now below the diagonal and all data points in the third quarter are above the diagonal, and the change in slope is monotonic across panels, as expected.

We observe some non-monotonicity across panels in the current Figure 3, that is, (^2^*) underestimates the observed rate of non-home fitness gains at low and the same issue as above. But there is also no expectation that2* should slightly overestimates it at higher values. This non-monotonicity is not caused by consistently over- or under-estimate the observed rates. At this point, we believe that the observed behavior is a property of the approximation derived by Good et al. (PNAS 2015).

Reviewer #3:The goal of this manuscript--to develop predictive tools for inferring fitness trajectories in new environments--is an important goal and I appreciate the synthesis of theoretical modeling with parameter estimation from empirical mutation studies.Reading through the manuscript, however, I found myself repeatedly wondering whether the stated application of the methods developed here doesn't constitute something of a tautology. This could be a misreading on my end, but I'll explain: the authors state that they have the central goal of predicting whether a population adapting to one environment will lose fitness in another "non-home" environment. Yet the parameter estimation they develop and propose for estimating fitness trajectories requires fitness measurements in both the home and non-home environments. If one already has fitness measurements for both home and non-home, how much more information is added by estimating the JDFE? I understand that the authors are estimating the fitness trajectories over time, with the incorporation of population genetic parameters, but again, I was unsure of how much information was added with the JDFE particularly given large discrepancies in the Wright-Fisher models and the decreasing predictive capacity with time. The bottom row of Figure 1 provided perhaps the most convincing evidence of the usefulness of the JDFE, but the unintuitive result was not adequately explored nor explained (see comment below). Also, perhaps an exploration of how the predictions could be extended to unmeasured environments is possible (as in Kinsler et al. 2020)?

There are several interrelated issues in this comment: (1) Large discrepancies between theory and simulations; (2) Is the theory tautological? (3) If one already has fitness measurements for both home and non-home, how much more information is added by estimating the JDFE? (4) Unintuitive result in Figure 1 is not explained (5) Can predictions be made about unmeasured environments? We now address these issues one by one.

1) This concern was also raised by Reviewer 1, and we address it in more detail in response to him. The main point is that we have extended our theory and now have good quantitative correspondence between theory and simulations.

2) Our theory is not tautological, which can be asserted on several accounts. Consider Figure 2 (former Figure 1), which is the idealized case when the full JDFE is known. The reviewer admits that even in this idealized case it is not obvious how to make a basic qualitative prediction: will an average population evolve collateral resistance or sensitivity? Prediction is challenging because different mutations contribute differently to adaptation, and a population genetic theory is necessary to properly weigh them. Our theory tells us what summary statistics of the JDFE we need to use to correctly predict the rate of collateral evolution (as well as the uncertainty around this expectation).

While predicting the average qualitative collateral outcome is already challenging, we argue that what really matters in practice is to rank drug pairs by their *risk* of collateral resistance. Assessing this risk seems impossible without a population genetic model. We have now added a new section where we show how our theory helps us rank drug pairs (section “Robust ranking of drug pairs”).

3) The reviewer alludes here to a more realistic scenario when the full JDFE is not known, but instead “one already has fitness measurements for both home and non-home” from a finite sample of mutants. The problems are the same as in the idealized case with a fully known JDFE: How do we combine these fitness measurements into an effective estimator of the rate of collateral evolution? If we have fitness measurements for these mutants in the presence of multiple drugs, how do we use these measurements to rank drug pairs according to the risk of collateral resistance? Again, our theory provides answers to these questions. Furthermore, we have now added a new section “Measuring JDFEs” where we discuss the issues related to measurement of JDFE.

4) Thank you for pointing out this omission. Our theory does indeed help us understand the observations made in Figure 1. We have added the relevant discussion (see p. 9, l. 245).

5) The question of how collateral evolution would proceed in unmeasured environments is fascinating and important, but unfortunately it cannot be answered within the current JDFE framework. To answer it, we either need to have some heuristic knowledge of how the effects of mutations vary across environments or we need to know something about the mechanistic basis of resistance and collateral resistance/sensitivity. Fortunately, a lot is known about these mechanisms, at least for some drug pairs, and we are now working in this direction.

Further specific conceptual comments and suggestions:1) The authors demonstrate in Figure 1 that JDFEs even with similar shapes produce markedly different fitness trajectories. They argue that the correlation coefficient of the JDFE is not a reliable predictor of fitness trajectories in the home environment. I was struck by this counterintuitive result, and found myself searching for further explanation. Are the authors arguing that the practice of simply looking at the correlation coefficient in tradeoff studies in general is insufficient for predicting the fates of pleiotropic mutations? Either way, it would be helpful to the reader to elaborate on why and under which conditions the discrepancy with the correlation coefficient and fitness trajectories arises.

Correct: correlation between the effects of mutations on traits is not sufficient to predict pleiotropic outcomes of evolution. As mentioned above, our theory helps us understand why this is so (see p. 9, l. 245).

2) The modeling results throughout the manuscript reveal poor predictive capabilities in Wright-Fisher simulations. For example, the results in figure 2 show substantial discrepancy between the theoretical predictions and the results of the Wright-Fisher simulations. The authors address this only briefly stating that outside of the strong selection, weak mutation model (SSWM) the pleiotropy statistics are only "statistical predictors". But the discrepancy was systematic and wide, suggesting rather little insight from the pleiotropy statistics in sequential adaptation scenarios. I could not find discussion of this discrepancy between the SSWM and Wright-Fisher modeling predictions.

As mentioned above, this is now resolved.

[Editors’ note: what follows is the authors’ response to the second round of review.]

Essential revisions:The manuscript has been improved considerably. The reviewers particularly appreciated the more general theoretical results. However, we would like to see a more thorough discussion of previous literature. In particular, the study by Nichol et al. is important prior work that needs to be discussed in greater depth.

We have now expanded our discussion of the prior literature in the Introduction (LL. 25–38). We also added a more extensive discussion of the Nichol et al. paper specifically (LL. 73–79).

In addition to a more extensive discussion of the literature, we would also like to see, if at all possible, some level of empirical validation of the results beyond the KO data presented so far. The data by Stiffler et al. and Nichol et al. characterize point mutations. Using these data beyond what is currently shown in Figure S1 could be very valuable.

We agree that we have not exploited the data by Stiffler et al. to its fullest extent in the previous version of the manuscript. We now utilize it further to show that our theory predicts the correct *in silico* ranking of drug pairs even for these more realistic non-Gaussian JDFEs. This new result, shown in the new Figure 4 —figure supplement 1, bolsters the optimism that our theory will work beyond *in silico* experiments. However, we realize that this is not an empirical validation and agree that such validation would be highly desirable.

An empirical validation of our theory would require two pieces of data: (1) an estimate of a JDFE for at least one pair of drugs and (2) an evolution experiment in the presence of one of these drugs as well as a fitness measurement after evolution in the presence of the other drug. Notably, the evolution experiment must be carried out with a sufficient number of replicate populations so that we can estimate the probability to evolve collateral resistance, or at least the variance in the collateral outcome. We therefore systematically searched the existing literature for datasets that satisfy these two criteria. We describe this meta-analysis below. The main conclusions of this meta-analysis are the following.

First, there is no single study that provides both pieces of data. Second, there are a handful of studies that can be paired so that one provides a JDFE estimate and the other provides experimental evolution data. Stiffler et al. and Nichol et al. form one such pair. However, none of the pairs are sufficiently aligned in terms of their experimental design that would allow us to test our theory without serious confounding factors. The biggest problem is that full-genome JDFEs have not yet been reported in the literature. JDFEs can be estimated for single genes from deep mutational scanning data. However, almost all evolution experiments are carried out with whole organisms and resistance mutations arise across several genes.

Thus, it is unfortunately not possible to empirically validate our theory using existing data. We plan to carry out the necessary experiments in our laboratory, but this work goes far beyond the scope of this paper.

Reviewer #3:The authors thoroughly responded to the reviewers’ comments and I found the resubmission to be both clearer and to demonstrate greater prediction accuracy in the Wright Fisher simulations. The addition of the section on estimating JDFE parameters from experimental data was a positive addition to the manuscript in that it provides a bridge for experimentalists to implement the methods developed by the authors.That being said, as an experimentalist who could potentially implement the proposed modeling in my own work for predicting tradeoffs, I am not yet convinced of the significant advance of the proposed modellng framework for making predictions. Specifically, I found the following two points to present the most significant drawbacks to the manuscript at present:i) I found the manuscript to lack sufficient discussion of what has been shown before in the field of modeling collateral resistances and how the present manuscript presents a clear advance in light of this work. To the first point, a brief perusal of recent literature on collateral resistance brought me to Nichol et al. 2019 Nature Comm. Ardell et al. reference the Nichol manuscript on line 37 when stating that previous work observes wide variation in collateral outcomes. But Nichol et al. did more than demonstrate variation in collateral outcomes, and instead conducted 60 parallel experimental evolution assays in one antibiotic, measured the probability of collateral resistance/susceptibility and then modeled through SSWM simulations the predicted collateral resistance outcomes for dozens of drug pairs. The present manuscript should explain how its methods/goals/results differ from those of Nichol et al.

We have expanded the introduction to address this concern, as described in the response to the editor. We agree that the paper by Nichol et al. is important and very relevant to the present study, as we now discuss in LL. 72–79 and mention in several other places in the text. In particular, Nichol et al. are motivated by the same problem as our work. However, our approach to solving it is conceptually different, as we discuss in LL. 79–98. It is also worth noting that Nichol et al. did not quantitatively test their model against their own experimental data. In fact, their experimental design does not allow them to do so, as it suffers from the same problem as we face in testing our theory, i.e., the fact that their theory is based on fitness landscapes consisting of four mutations in a single gene, but resistance mutations in their experiment often arise elsewhere in the genome. In the absence of direct empirical tests, we argue on theoretical grounds that the JDFE-based approach is superior to the approach based on combinatorically complete landscapes (LL. 79–98).

My second point is (ii) the manuscript would be significantly strengthened if it could provide proof-of-concept validations beyond the KO work and the β-lactamase work. If I understand correctly, the authors perform the drug-ranking experiments with simulated data. I am surprised that the authors cannot find a dataset in which to validate any part of the drug-ranking predictions. This type of validation would be helpful in convincing the reader of the strength of the proposed methods. As a relevant aside, Beyond Figure S1 I couldn't find where the Β-lactamase data was used and the basic conclusion stated in the text for S1 regarding variable resistance pleiotropy is already well-established in previous work.

Please see our detailed response to the editor’s comment above, as well as the results of our systematic literature search described below. In short, we agree that empirical validation is important. However, it is unfortunately impossible to validate our theory with existing data. This may be surprising given the vastness of the literature on antibiotic resistance, but this lack of relevant data can perhaps be explained in hindsight. First, the technical capabilities for estimating genome-wide JDFE (e.g., using the barcode-lineage tracking method) have appeared only recently and are not yet widely adopted. And second, the importance of stochasticity for collateral outcomes is not yet widely appreciated in the field. We hope that the reviewer appreciates the possibility that publishing this theoretical study may in fact stimulate empiricists to carry out the relevant measurements.